# Resonating neurons stabilize heterogeneous grid-cell networks

**Divyansh Mittal, Rishikesh Narayanan\***

Cellular Neurophysiology Laboratory, Molecular Biophysics Unit, Indian Institute of Science, Bangalore, India

**Abstract** A central theme that governs the functional design of biological networks is their ability to sustain stable function despite widespread parametric variability. Here, we investigated the impact of distinct forms of biological heterogeneities on the stability of a two-dimensional continuous attractor network (CAN) implicated in grid-patterned activity generation. We show that increasing degrees of biological heterogeneities progressively disrupted the emergence of grid-patterned activity and resulted in progressively large perturbations in low-frequency neural activity. We postulated that targeted suppression of low-frequency perturbations could ameliorate heterogeneity-induced disruptions of grid-patterned activity. To test this, we introduced intrinsic resonance, a physiological mechanism to suppress low-frequency activity, either by adding an additional high-pass filter (phenomenological) or by incorporating a slow negative feedback loop (mechanistic) into our model neurons. Strikingly, CAN models with resonating neurons were resilient to the incorporation of heterogeneities and exhibited stable grid-patterned firing. We found CAN models with mechanistic resonators to be more effective in targeted suppression of low-frequency activity, with the slow kinetics of the negative feedback loop essential in stabilizing these networks. As low-frequency perturbations ($1/f$ noise) are pervasive across biological systems, our analyses suggest a universal role for mechanisms that suppress low-frequency activity in stabilizing heterogeneous biological networks.

**\*For correspondence:**
rishi@iisc.ac.in

**Competing interests:** The authors declare that no competing interests exist.

## Introduction

Stability of network function, defined as the network's ability to elicit robust functional outcomes despite perturbations to or widespread variability in its constitutive components, is a central theme that governs the functional design of several biological networks. Biological systems exhibit ubiquitous parametric variability spanning different scales of organization, quantified through statistical heterogeneities in the underlying parameters. Strikingly, in spite of such large-scale heterogeneities, outputs of biological networks are stable and are precisely tuned to meet physiological demands. A central question that spans different scales of organization is on the ability of biological networks to achieve physiological stability in the face of ubiquitous parametric variability (*Turrigiano and Nelson, 2000*; *Edelman and Gally, 2001*; *Maslov and Sneppen, 2002*; *Stelling et al., 2004*; *Marder and Goaillard, 2006*; *Barkai and Shilo, 2007*; *Kitano, 2007*; *Félix and Barkoulas, 2015*).

Biological heterogeneities are known to play critical roles in governing stability of network function, through intricate and complex interactions among mechanisms underlying functional emergence (*Edelman and Gally, 2001*; *Renart et al., 2003*; *Marder and Goaillard, 2006*; *Tikidji-Hamburyan et al., 2015*; *Mishra and Narayanan, 2019*; *Rathour and Narayanan, 2019*). However, an overwhelming majority of theoretical and modeling frameworks lack the foundation to evaluate the impact of such heterogeneities on network output, as they employ unnatural homogeneous networks in assessing network function. The paucity of heterogeneous network frameworks is partly attributable to the enormous analytical or computational costs involved in assessing heterogeneous networks. In this study, we quantitatively address questions on the impact of distinct forms of

biological heterogeneities on the functional stability of a two-dimensional continuous attractor network (CAN), which has been implicated in the generation of patterned neuronal activity in grid cells of the medial entorhinal cortex (*Burak and Fiete, 2009*; *Knierim and Zhang, 2012*; *Couey et al., 2013*; *Domnisoru et al., 2013*; *Schmidt-Hieber and Häusser, 2013*; *Yoon et al., 2013*; *Tukker et al., 2021*). Although the continuous attractor framework has offered insights about information encoding across several neural circuits (*Samsonovich and McNaughton, 1997*; *Seung et al., 2000*; *Renart et al., 2003*; *Wills et al., 2005*; *Burak and Fiete, 2009*; *Knierim and Zhang, 2012*; *Schmidt-Hieber and Häusser, 2013*; *Yoon et al., 2013*; *Kim et al., 2017*), the fundamental question on the stability of 2D CAN models in the presence of biological heterogeneities remains unexplored. Here, we systematically assessed the impact of biological heterogeneities on stability of emergent spatial representations in a 2D CAN model and unveiled a physiologically plausible neural mechanism that promotes stability despite the expression of heterogeneities.

We first developed an algorithm to generate virtual trajectories that closely mimicked animal traversals in an open arena, to provide better computational efficiency in terms of covering the entire arena within shorter time duration. We employed these virtual trajectories to drive a rate-based homogeneous CAN model that elicited grid-patterned neural activity (*Burak and Fiete, 2009*) and systematically introduced different degrees of three distinct forms of biological heterogeneities. The three distinct forms of biological heterogeneities that we introduced, either individually or together, were in neuronal intrinsic properties, in afferent inputs carrying behavioral information and in local-circuit synaptic connectivity. We found that the incorporation of these different forms of biological heterogeneities disrupted the emergence of grid-patterned activity by introducing perturbations in neural activity, predominantly in low-frequency components. In the default model where neurons were integrators, grid patterns and spatial information in neural activity were progressively lost with increasing degrees of biological heterogeneities and were accompanied by progressive increases in low-frequency perturbations.

As heterogeneity-induced perturbations to neural activity were predominantly in the lower frequencies, we postulated that suppressing low-frequency perturbations could ameliorate the disruptive impact of biological heterogeneities on grid-patterned activity. We recognized intrinsic neuronal resonance as an established biological mechanism that suppresses low-frequency components, effectuated by the expression of resonating conductances endowed with specific biophysical characteristics (*Hutcheon and Yarom, 2000*; *Narayanan and Johnston, 2008*). Consequently, we hypothesized that intrinsic neuronal resonance could stabilize the heterogeneous grid-cell network through targetted suppression of low-frequency perturbations. To test this hypothesis, we developed two distinct strategies to introduce intrinsic resonance in our rate-based neuronal model to mimic the function of resonating conductances in biological neurons: (1) a *phenomenological* approach where an additional tunable high-pass filter (HPF) was incorporated into single-neuron dynamics and (2) a *mechanistic* approach where resonance was realized through a slow negative feedback loop akin to the physiological mechanism behind neuronal intrinsic resonance (*Hutcheon and Yarom, 2000*). We confirmed that the emergence of grid-patterned activity was not affected by replacing all the integrator neurons in the homogeneous CAN model with theta-frequency resonators (either phenomenological or mechanistic). We systematically incorporated different forms of biological heterogeneities into the 2D resonator CAN model and found that intrinsic neuronal resonance stabilized heterogeneous neural networks, through suppression of low-frequency components of neural activity. Although this stabilization was observed with both phenomenological and mechanistic resonator networks, the mechanistic resonator was extremely effective in suppressing low-frequency activity without introducing spurious high-frequency components into neural activity. Importantly, we found that the slow kinetics of the negative feedback loop was essential in stabilizing CAN networks built with mechanistic resonators.

Together, our study unveils an important role for intrinsic neuronal resonance in stabilizing network physiology through the suppression of heterogeneity-induced perturbations in low-frequency components of network activity. Our analyses suggest that intrinsic neuronal resonance constitutes a cellular-scale activity-dependent negative feedback mechanism, a specific instance of a well-established network motif that effectuates stability and suppresses perturbations across different biological networks (*Savageau, 1974*; *Becskei and Serrano, 2000*; *Thattai and van Oudenaarden, 2001*; *Austin et al., 2006*; *Dublanche et al., 2006*; *Raj and van Oudenaarden, 2008*; *Lestas et al., 2010*; *Cheong et al., 2011*; *Voliotis et al., 2014*). As the dominance of low-frequency perturbations

is pervasive across biological networks (*Hausdorff and Peng, 1996*; *Gilden, 2001*; *Gisiger, 2001*; *Ward, 2001*; *Buzsaki, 2006*), we postulate that mechanisms that suppress low-frequency components could be a generalized route to stabilize heterogeneous biological networks.

## Results

The rate-based CAN model consisting of a 2D neural sheet (default size: 60 × 60 = 3600 neurons) with the default integrator neurons for eliciting grid-cell activity was adopted from *Burak and Fiete, 2009*. The sides of this neural sheet were connected, yielding a toroidal (or periodic) network configuration. Each neuron in the CAN model received two distinct sets of synaptic inputs, one from other neurons within the network and another feed-forward afferent input that was dependent on the velocity of the virtual animal. The movement of the virtual animal was modeled to occur in a circular 2D spatial arena (*Figure 1A; Real trajectory*) and was simulated employing recordings of rat movement (*Hafting et al., 2005*) to replicate earlier results (*Burak and Fiete, 2009*). However, the real trajectory spanned 590 s of real-world time and required considerable simulation time to sufficiently explore the entire arena, essential for computing high-resolution spatial activity maps. Consequently, the computational costs required for exploring the parametric space of the CAN model, with different forms of network heterogeneities in networks of different sizes and endowed with distinct kinds of neurons, were exorbitant. Therefore, we developed a virtual trajectory, mimicking smooth animal movement within the arena, but with sharp turns at the borders.

Our analyses demonstrated the ability of virtual trajectories to cover the entire arena with lesser time (*Figure 1A*; Virtual trajectory), while not compromising on the accuracy of the spatial activity maps constructed from this trajectory in comparison to the maps obtained with the real trajectory (*Figure 1B*, *Figure 1—figure supplement 1*). Specifically, the correlation values between the spatial autocorrelation of rate maps of individual grid cells for real trajectory and that from the virtual trajectory at different rat runtimes ($T_{run}$) were high across all measured runtimes (*Figure 1B*). These correlation values showed a saturating increase with increase in $T_{run}$. As there was no substantial improvement in accuracy beyond $T_{run}$ = 100 s, we employed $T_{run}$ = 100 s for all simulations (compared to the 590 s of the real trajectory). Therefore, our virtual trajectory covered similar areas in approximately six times lesser duration (*Figure 1A*), which reduced the simulation duration by a factor of ~10 times when compared to the real rat trajectory, while yielding similar spatial activity maps (*Figure 1B*, *Figure 1—figure supplement 1*). Our algorithm also allowed us to generate fast virtual trajectories mimicking rat trajectories in open arenas of different shapes (Circle: *Figure 1A*; Square: *Figure 2E*).

### Biologically prevalent network heterogeneities disrupted the emergence of grid-cell activity in CAN models

The simulated CAN model (*Figure 1*) is an idealized homogeneous network of intrinsically identical neurons, endowed with precise local connectivity patterns and identical response properties for afferent inputs. Although this idealized CAN model provides an effective phenomenological framework to understand and simulate grid-cell activity, the underlying network does not account for the ubiquitous biological heterogeneities that span neuronal intrinsic properties and synaptic connectivity patterns. Would the CAN model sustain grid-cell activity in a network endowed with different degrees of biological heterogeneities in neuronal intrinsic properties and strengths of afferent/local synaptic inputs?

To address this, we systematically introduced three distinct forms of biological heterogeneities into the rate-based CAN model. We introduced *intrinsic heterogeneity* by randomizing the value of the neuronal integration time constant $\tau$ across neurons in the network. Within the rate-based framework employed here, randomization of $\tau$ reflected physiological heterogeneities in neuronal excitability properties, and larger spans of $\tau$ defined higher degrees of intrinsic heterogeneity (*Table 1*; *Figure 2A*). *Afferent heterogeneity* referred to heterogeneities in the coupling of afferent velocity inputs onto individual neurons. Different degrees of afferent heterogeneities were introduced by randomizing the velocity scaling factor ($\alpha$) in each neuron through uniform distributions of different ranges (*Table 1*; *Figure 2B*). S*ynaptic heterogeneity* involved the local connectivity matrix, and was introduced as additive jitter to the default center-surround connectivity matrix. Synaptic jitter was independently sampled for each connection, from a uniform distribution whose differential

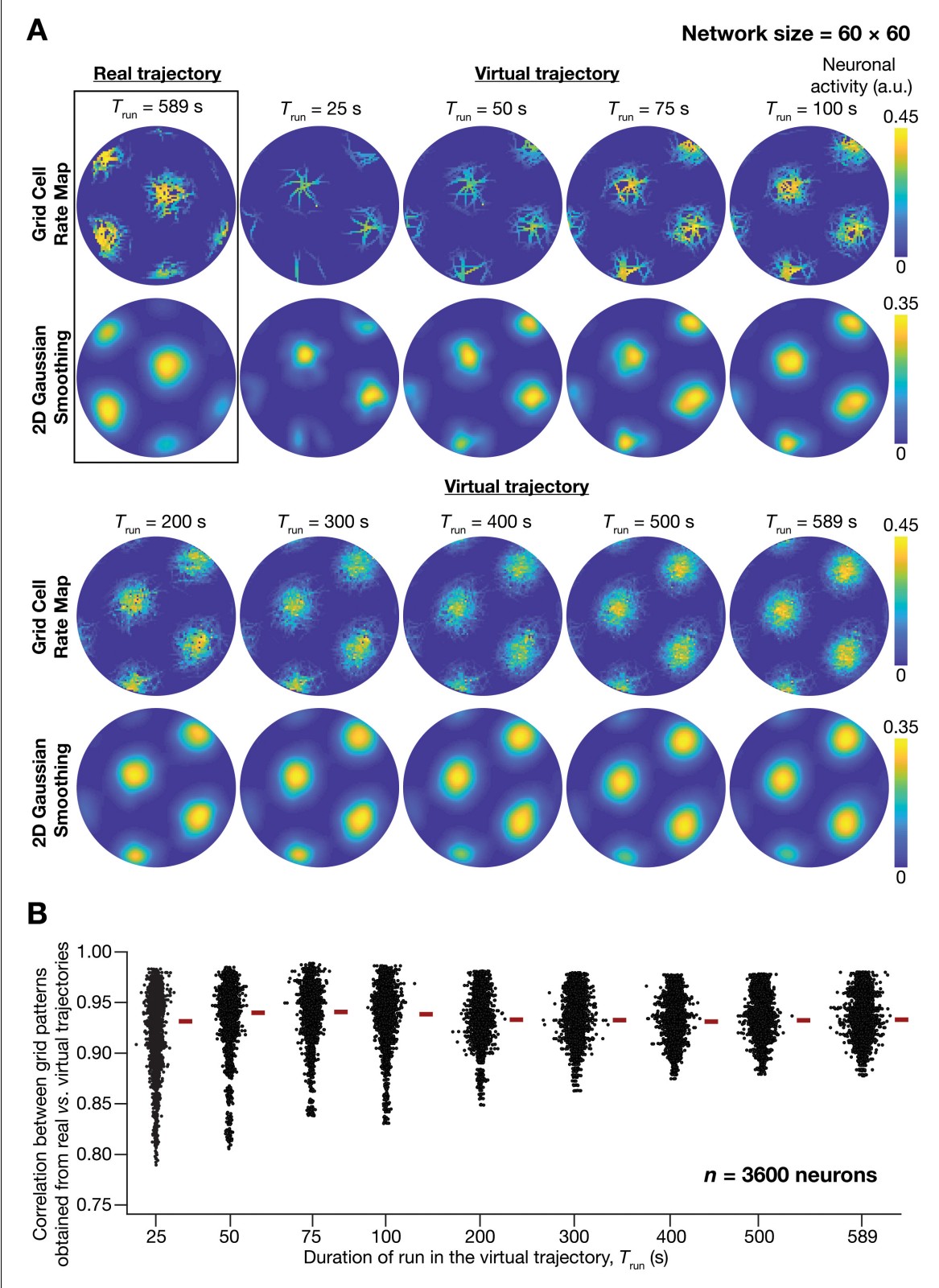

**Figure 1.** A fast virtual trajectory developed for simulating rodent run in a two-dimensional circular arena elicits grid-cell activity in a continuous attractor network (CAN) model. (**A**) *Panels within rectangular box*: simulation of a CAN model (60 × 60 neural network) using a 589 s long real trajectory from a rat (***Hafting et al., 2005***) yielded grid-cell activity. *Other panels*: A virtual trajectory (see Materials and methods) was employed for simulating a CAN model (60 × 60 neural network) for different time points. The emergent activity patterns for nine different run times ($T_{run}$) of the virtual animal are

*Figure 1 continued on next page*

*Figure 1 continued*

shown to yield grid-cell activity. Top subpanels show color-coded neural activity through the trajectory, and bottom subpanels represent a smoothened spatial profile of neuronal activity shown in the respective top subpanels. (B) Pearson's correlation coefficient between the spatial autocorrelation of rate maps using the real trajectory and the spatial autocorrelation of rate maps from the virtual trajectory for the nine different values of $T_{run}$, plotted for all neurons (n = 3600) in the network.

The online version of this article includes the following figure supplement(s) for figure 1:

**Figure supplement 1.** Quantitative comparison of grid-cell firing properties obtained from CAN models simulated with virtual vs. real trajectories.

span regulated the associated higher degree of heterogeneity (*Table 1*; *Figure 2C*). In homogeneous networks, $\tau$ (=10 ms), $\alpha$ (=45) and $W_{ij}$ (center-shifted Mexican hat connectivity without jitter) were identical for all neurons in the network. We assessed four distinct sets of heterogeneous networks: three sets endowed with one of intrinsic, afferent and synaptic heterogeneities, and a fourth where all three forms of heterogeneities were co-expressed. When all heterogeneities were present together, all three sets of parameters were set randomly with the degree of heterogeneity defining the bounds of the associated distribution (*Table 1*).

We found that the incorporation of any of the three forms of heterogeneities into the CAN model resulted in the disruption of grid pattern formation, with the deleterious impact increasing with increasing degree of heterogeneity (*Figures 2E* and *3*). Quantitatively, we employed grid score (*Fyhn et al., 2004*; *Hafting et al., 2005*) to measure the emergence of pattern formation in the CAN model activity, and found a reduction in grid score with increase in the degree of each form of heterogeneity (*Figure 2F*). We found a hierarchy in the disruptive impact of different types of heterogeneities, with synaptic heterogeneity producing the largest reduction to grid score, followed by afferent heterogeneity. Intrinsic heterogeneity was the least disruptive in the emergence of grid-cell activity, whereby a high degree of heterogeneity was required to hamper grid pattern formation (*Figure 2E,F*). Simultaneous progressive introduction of all three forms of heterogeneities at multiple degrees resulted in a similar progression of grid-pattern disruption (*Figure 2E,F*). The introduction of the highest degree of all heterogeneities into the CAN model resulted in a complete loss of patterned activity and a marked reduction in the grid score values of all neurons in the network (*Figure 2E,F*). We confirmed that these results were not artifacts of specific network initialization choices by observing similar grid score reductions in five additional trials with distinct initializations of CAN models endowed with all three forms of heterogeneities (*Figure 3—figure supplement 1F*). In addition, to rule out the possibility that these conclusions were specific to the choice of the virtual trajectory employed, we repeated our simulations and analyses with a different virtual trajectory (*Figure 3—figure supplement 2A*) and found similar reductions in grid score with increase in the degree of all heterogeneities (*Figure 3—figure supplement 2B*).

Detailed quantitative analysis of grid-cell activity showed that the introduction of heterogeneities did not result in marked population-wide changes to broad measures such as average firing rate, mean size of grid fields and average grid spacing (*Figure 3*, *Figure 3—figure supplements 1–2*). Instead, the introduction of parametric heterogeneities resulted in a loss of spatial selectivity and disruption of patterned activity, reflected as progressive reductions in grid score, peak firing rate, information rate and sparsity (*Figures 2–3*, *Figure 3—figure supplements 1–2*). Our results showed that in networks with intrinsic or afferent heterogeneities, grid score of individual neurons was not dependent on the actual value of the associated parameter ($\tau$ or $\alpha$, respectively), but was critically reliant on the degree of heterogeneity (*Figure 3H*). Specifically, for a given degree of intrinsic or afferent heterogeneity, the grid score value spanned similar ranges for the low or high value of the associated parameter; but grid score reduced with increased degree of heterogeneities. In addition, grid score of cells reduced with increase in local synaptic jitter, which increased with the degree of synaptic heterogeneity (*Figure 3H*).

Thus far, our analyses involved a CAN model with a specific size (60×60). Would our results on heterogeneity-driven disruption of grid-patterned activity extend to CAN models of other sizes? To address this, we repeated our simulations with networks of size 40×40, 50×50, 80×80 and 120×120, and introduced different degrees of all three forms of heterogeneities into the network. We found that the disruptive impact of heterogeneities on grid-patterned activity was consistent across all tested networks (*Figure 3—figure supplement 3A*), manifesting as a reduction in grid

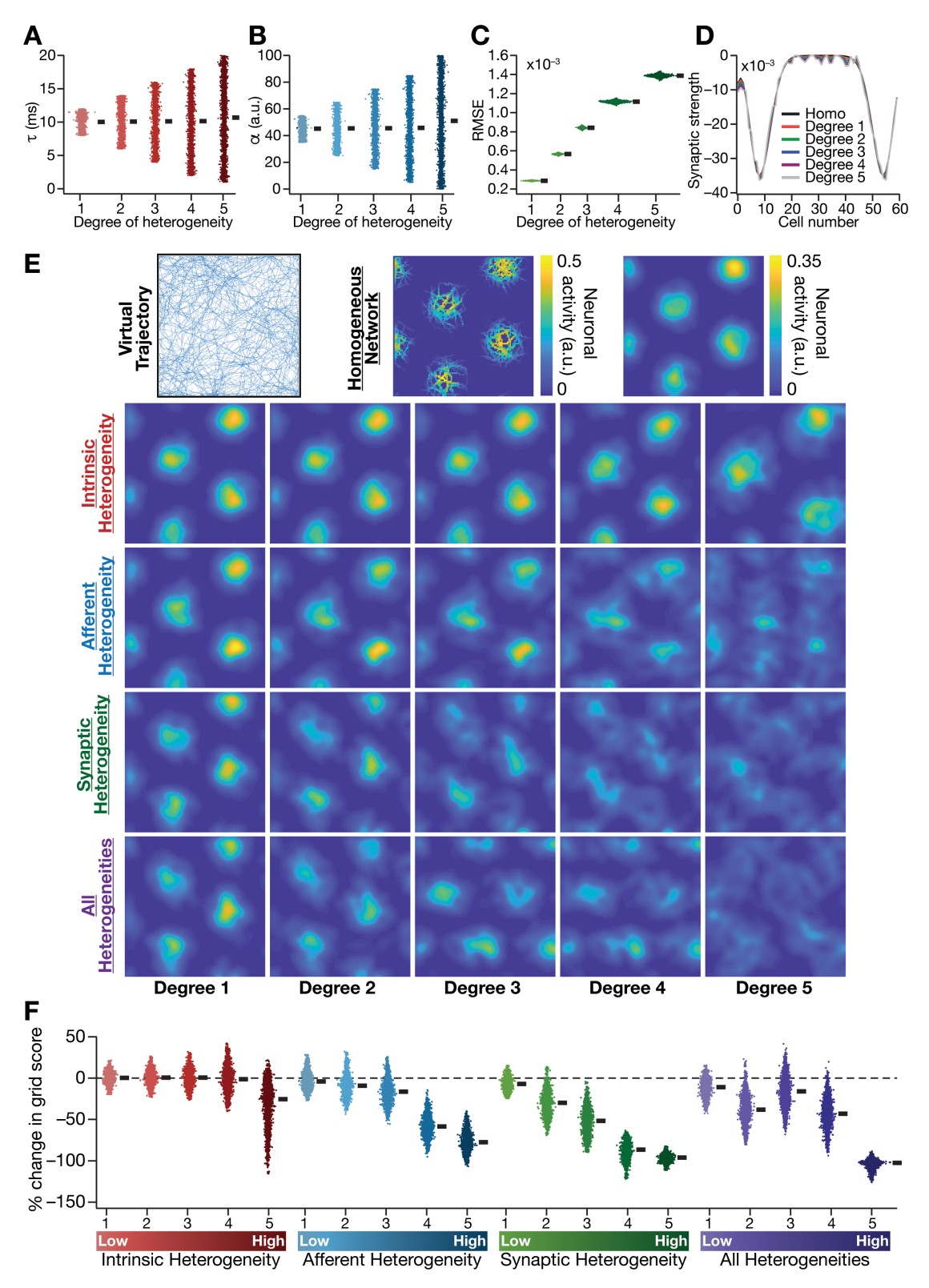

**Figure 2.** Biologically prevalent network heterogeneities disrupt the emergence of grid-cell activity in CAN models. (**A**) Intrinsic heterogeneity was introduced by setting the integration time constant ($\tau$) of each neuron to a random value picked from a uniform distribution, whose range was increased to enhance the degree of intrinsic heterogeneity. The values of $\tau$ for the 3600 neurons in the 60 × 60 CAN model are shown for 5 degrees of intrinsic heterogeneity. (**B**) Afferent heterogeneity was introduced by setting the velocity-scaling factor ($\alpha$) of each neuron to a random value picked

*Figure 2 continued on next page*

*Figure 2 continued*

from a uniform distribution, whose range was increased to enhance the degree of intrinsic heterogeneity. The values of $\alpha$ for the 3600 neurons are shown for 5 degrees of afferent heterogeneity. (C) Synaptic heterogeneity was introduced as an additive jitter to the intra-network Mexican hat connectivity matrix, with the magnitude of additive jitter defining the degree of synaptic heterogeneity. Plotted are the root mean square error (RMSE) values computed between the connectivity matrix of 'no jitter' case and that for different degrees of synaptic heterogeneities, across all synapses. (D) Illustration of a one-dimensional slice of synaptic strengths with different degrees of heterogeneity, depicted for Mexican-hat connectivity of a given cell to 60 other cells in the network. (E) *Top left*, virtual trajectory employed to obtain activity patterns of the CAN model. *Top center*, Example rate maps of grid-cell activity in a homogeneous CAN model. *Top right*, smoothed version of the rate map. *Rows 2–4*: smoothed version of rate maps obtained from CAN models endowed with five different degrees (increasing left to right) of disparate forms (*Row 2*: intrinsic; *Row 3*: afferent; *Row 4*: synaptic; and *Row 5*: all three heterogeneities together) of heterogeneities. (F) Percentage change in the grid score of individual neurons (n = 3600) in networks endowed with the four forms and five degrees of heterogeneities, compared to the grid score of respective neurons in the homogeneous network.

score with increased degree of heterogeneities (*Figure 3—figure supplement 3B*). Our results also showed that the disruptive impact of heterogeneities on grid-patterned activity increased with increase in network size. This size dependence specifically manifested as a complete loss of spatially selective firing patterns (*Figure 3—figure supplement 3A*) and the consistent drop in the grid score value to zero (*i.e.*, %change = –100 in *Figure 3—figure supplement 3B*) across all neurons in the 120 × 120 network with high degree of heterogeneities. We also observed a progressive reduction in the average and the peak firing rates with increase in the degree of heterogeneities across all network sizes, although with a pronounced impact in networks with higher size (*Figure 3—figure supplement 3C,D*).

Together, our results demonstrated that the introduction of physiologically relevant heterogeneities into the CAN model resulted in a marked disruption of grid-patterned activity. The disruption manifested as a loss in spatial selectivity in the activity of individual cells in the network, progressively linked to the degree of heterogeneity. Although all forms of heterogeneities resulted in disruption of patterned activity, there was differential sensitivity to different heterogeneities, with heterogeneity in local network connectivity playing a dominant role in hampering spatial selectivity and *patterned* activity.

## Incorporation of biological heterogeneities predominantly altered neural activity in low frequencies

How does the presence of heterogeneities affect activity patterns of neurons in the network as they respond to the movement of the virtual animal? A systematic way to explore patterns of neural activity is to assess the relative power of specific frequency components in neural activity. Therefore, we subjected the temporal outputs of individual neurons (across the entire period of the simulation) to spectral analysis and found neural activity to follow a typical response curve that was dominated by lower frequency components, with little power in the higher frequencies (e.g., *Figure 4A*, HN). This is to be expected as a consequence of the low-pass filtering inherent to the neuronal model, reflective of membrane filtering.

We performed spectral analyses of temporal activity patterns of neurons from networks endowed with different forms of heterogeneities at various degrees (*Figure 4*). As biological heterogeneities were incorporated either by reducing or by increasing default parameter values (*Table 1*), there was considerable variability in how individual neurons responded to such heterogeneities (*Figure 4*).

**Table 1.** Forms and degrees of heterogeneities introduced in the CAN model of grid cell activity.

| Degree of heterogeneity | Intrinsic heterogeneity ($\tau$) | | Afferent heterogeneity ($\alpha$) | | Synaptic heterogeneity ($W_{ij}$) | |
|---|---|---|---|---|---|---|
| | Lower bound | Upper bound | Lower bound | Upper bound | Lower bound | Upper bound |
| 1 | 8 | 12 | 35 | 55 | 0 | 300 |
| 2 | 6 | 14 | 25 | 65 | 0 | 600 |
| 3 | 4 | 16 | 15 | 75 | 0 | 900 |
| 4 | 2 | 18 | 5 | 85 | 0 | 1200 |
| 5 | 1 | 20 | 0 | 100 | 0 | 1500 |

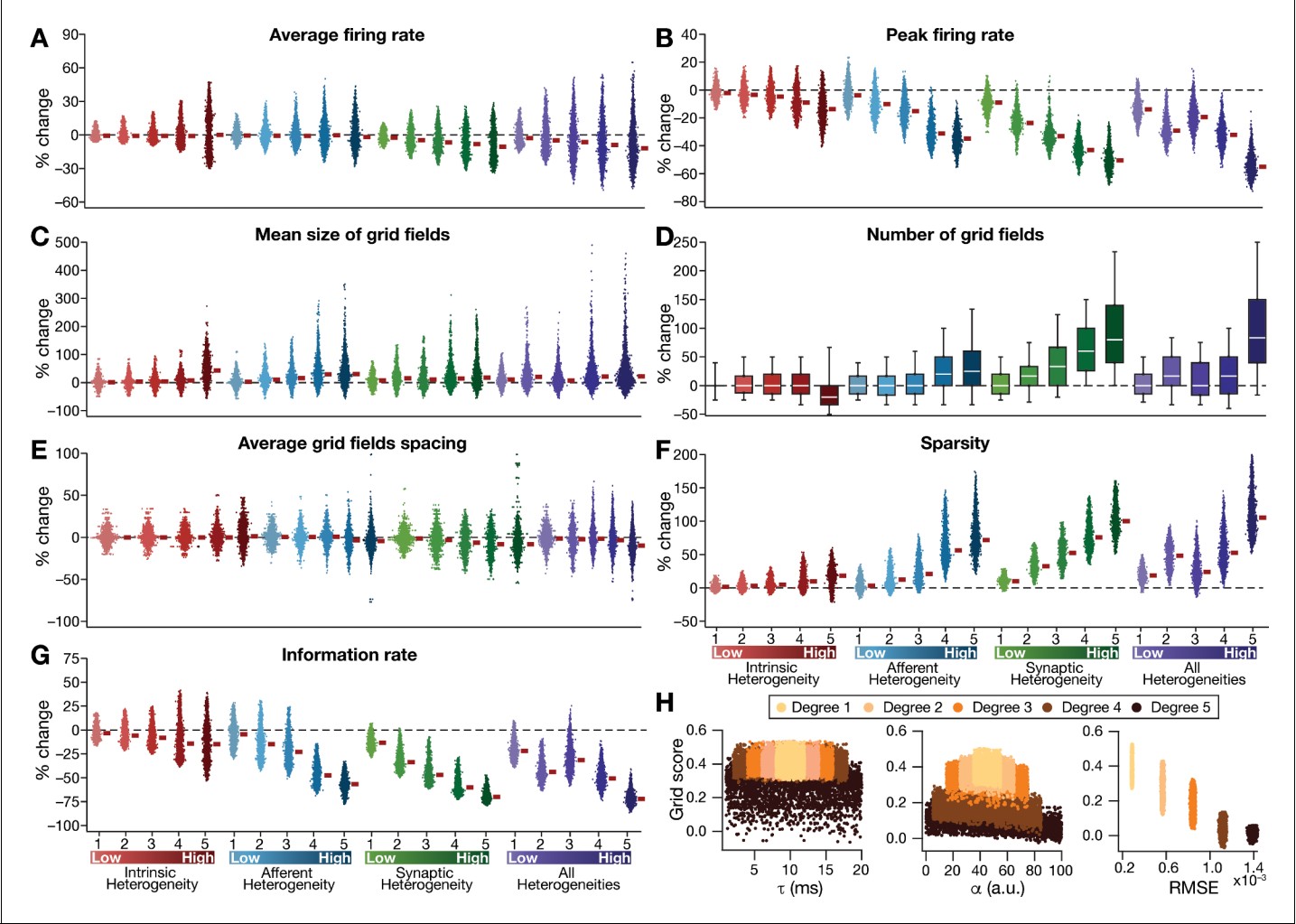

**Figure 3.** Quantification of the disruption in grid-cell activity induced by different forms of network heterogeneities in the CAN model. Grid-cell activity of individual neurons in the network was quantified by eight different measurements, for CAN models endowed independently with intrinsic, afferent, or synaptic heterogeneities or a combination of all three heterogeneities. (A–G) Depicted are percentage changes in each of average firing rate (A), peak firing rate (B), mean size (C), number (D), average spacing (E), information rate (F), and sparsity (G) for individual neurons (n = 3600) in networks endowed with distinct forms of heterogeneities, compared to the grid score of respective neurons in the homogeneous network. (H) Grid score of individual cells (n = 3600) in the network plotted as functions of integration time constant ($\tau$), velocity modulation factor ($\alpha$), and root mean square error (RMSE) between the connectivity matrices of the homogeneous and the heterogeneous CAN models. Different colors specify different degrees of heterogeneity. The three plots with reference to $\tau$, $\alpha$, and RMSE are from networks endowed with intrinsic, afferent, and synaptic heterogeneity, respectively.

The online version of this article includes the following figure supplement(s) for figure 3:

**Figure supplement 1.** Quantification of the disruption of grid-cell firing by network heterogeneities across different trials of CAN-model simulations.

**Figure supplement 2.** Disruption of grid-cell firing by network heterogeneities was invariant to the specific trajectory employed by the CAN models.

**Figure supplement 3.** Disruption of grid-cell activity by network heterogeneities was prevalent across CAN models of different sizes.

Specifically, when compared against the respective neuron in the homogeneous network, some neurons showed increases in activity magnitude at certain frequencies and others showed a reduction in magnitude (e.g., *Figure 4A*). To quantify this variability in neuronal responses, we first computed the difference between frequency-dependent activity profiles of individual neurons obtained in the presence of specific heterogeneities and of the same neuron in the homogeneous network (with identical initial conditions and subjected to identical afferent inputs). We computed activity differences for all the 3600 neurons in the network and plotted the variance of these differences as a function of frequency (e.g., *Figure 4B*). Strikingly, we found that the impact of introducing biological

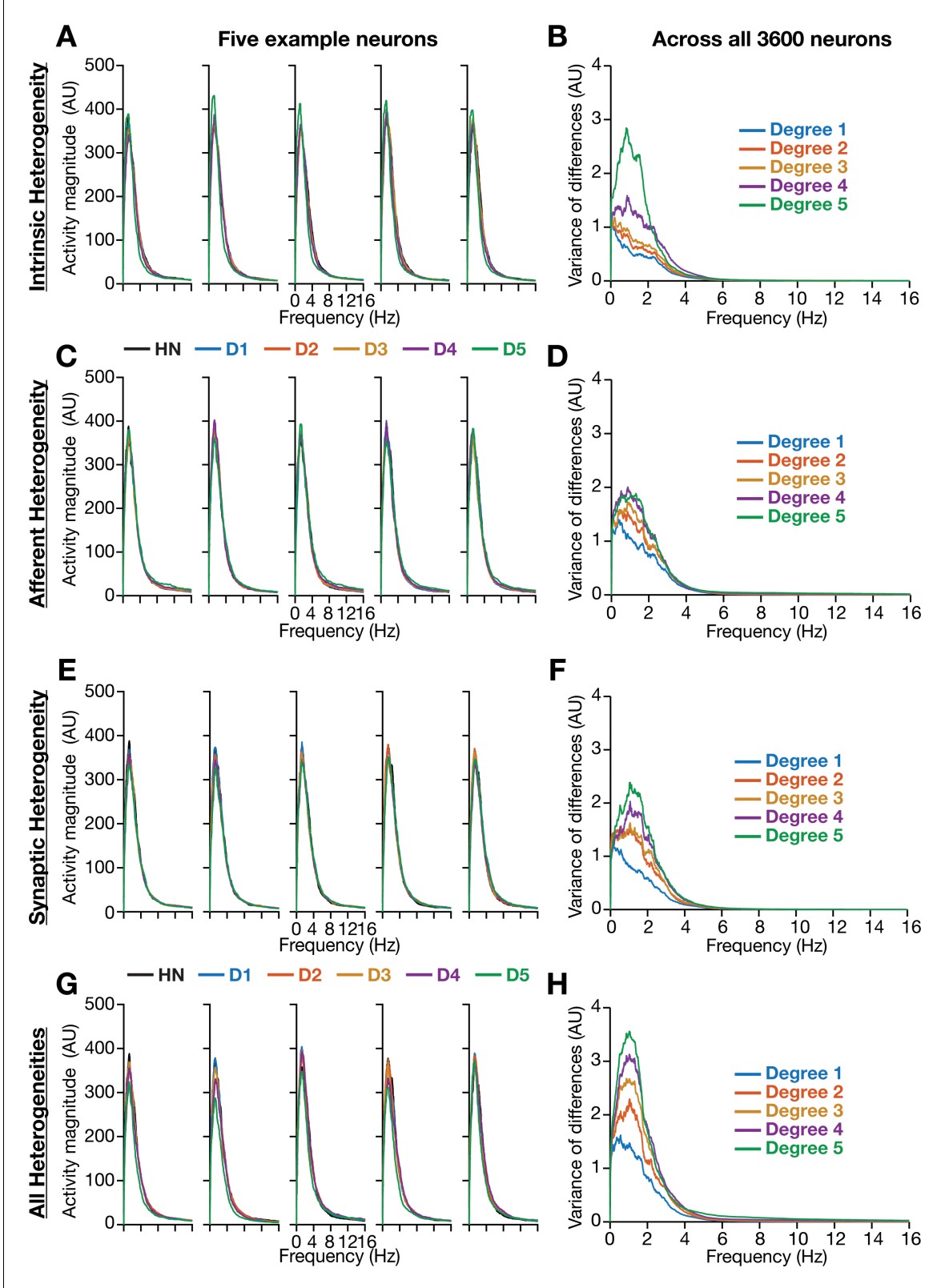

**Figure 4.** Incorporation of biological heterogeneities predominantly altered neural activity in low frequencies. (**A–H**) *Left,* Magnitude spectra of temporal activity patterns of five example neurons residing in a homogeneous network (HN) or in networks with different forms and degrees of heterogeneities. *Right,* Normalized variance of the differences between the magnitude spectra of temporal activity of neurons in homogeneous vs. heterogeneous networks, across different forms and degrees of heterogeneities, plotted as a function of frequency.

heterogeneities predominantly altered lower frequency components of neural activity, with little to no impact on higher frequencies. In addition, the variance in the deviation of neural activity from the homogenous network progressively increased with higher degrees of heterogeneities, with large deviations occurring in the lower frequencies (*Figure 4*).

Together, these observations provided an important insight that the presence of physiologically relevant network heterogeneities predominantly affected lower frequency components of neural activity, with higher degrees of heterogeneity yielding larger variance in the deviation of low-frequency activity from the homogeneous network.

## Introducing intrinsic resonance in rate-based neurons: phenomenological model

Several cortical and hippocampal neuronal structures exhibit intrinsic resonance, which allows these structures to maximally respond to a specific input frequency, with neural response falling on either side of this resonance frequency (*Hutcheon and Yarom, 2000*; *Narayanan and Johnston, 2007*; *Narayanan and Johnston, 2008*; *Das et al., 2017*). More specifically, excitatory (*Erchova et al., 2004*; *Giocomo et al., 2007*; *Nolan et al., 2007*; *Garden et al., 2008*; *Pastoll et al., 2012*) and inhibitory neurons (*Boehlen et al., 2016*) in the superficial layers of the medial entorhinal cortex manifest resonance in the theta frequency range (4–10 Hz). Therefore, the model of individual neurons in the CAN model as integrators of afferent activity is inconsistent with the resonating structure intrinsic to their physiological counterparts. Intrinsic resonance in neurons is mediated by the expression of slow restorative ion channels, such as the HCN or the *M*-type potassium, which mediate resonating conductances that suppress low-frequency inputs by virtue of their kinetics and voltage-dependent properties (*Hutcheon and Yarom, 2000*; *Narayanan and Johnston, 2008*; *Hu et al., 2009*). As the incorporation of biological heterogeneities predominantly altered low-frequency neural activity (*Figure 4*), we hypothesized that the expression of intrinsic neuronal resonance (especially the associated suppression of low-frequency components) could counteract the disruptive impact of biological heterogeneities on network activity, thereby stabilizing grid-like activity patterns.

An essential requirement in testing this hypothesis was to introduce intrinsic resonance in the rate-based neuronal models in our network. To do this, we noted that the integrative properties of the integrator model neuron are mediated by the low-pass filtering kinetics associated with the parameter $\tau$. We confirmed the low-pass filter (LPF) characteristics of the integrator neurons by recording the response of individual neurons to a chirp stimulus (*Figure 5A,B*). As this provides an equivalent to the LPF associated with the membrane time constant, we needed a HPF to mimic resonating conductances that suppress low-frequency activity in biological neurons. A simple approach to suppress low-frequency components is to introduce a differentiator, which we confirmed by passing a chirp stimulus through a differentiator (*Figure 5A,C*). We therefore passed the outcome of the integrator (endowed with LPF characteristics) through a first-order differentiator (HPF), with the postulate that the net transfer function would manifest resonance. We tested this by first subjecting a chirp stimulus to the integrator neuron dynamics and feeding that output to the differentiator. We found that the net output expressed resonance, acting as a band-pass filter (*Figure 5A,C*).

Physiologically, tuning of intrinsic resonance to specific frequencies could be achieved by altering the characteristics of the HPF or the LPF (*Hutcheon and Yarom, 2000*; *Narayanan and Johnston, 2008*; *Das et al., 2017*; *Mittal and Narayanan, 2018*; *Rathour and Narayanan, 2019*). Matching this physiological tuning procedure, we tuned resonance frequency ($f_{\mathrm{R}}$) in our rate-based resonator model either by changing $\tau$ that governs the LPF (*Figure 5D,E*) or by altering an exponent $\varepsilon$ (*Equation 9*) that regulated the slope of the HPF on the frequency axis (*Figure 5F–G*). We found $f_{\mathrm{R}}$ to decrease with an increase in $\tau$ (*Figure 5E*) or a reduction in $\varepsilon$ (*Figure 5G*). In summary, we mimicked the biophysical mechanisms governing resonance in neural structures to develop a phenomenological methodology to introduce and tune intrinsic resonance, yielding a tunable band-pass filtering structure in rate-based model neurons. In this model, as resonance was introduced through a HPF, a formulation that does not follow the mechanistic basis of resonance in physiological systems, we refer this as a *phenomenological model for resonating neurons*.

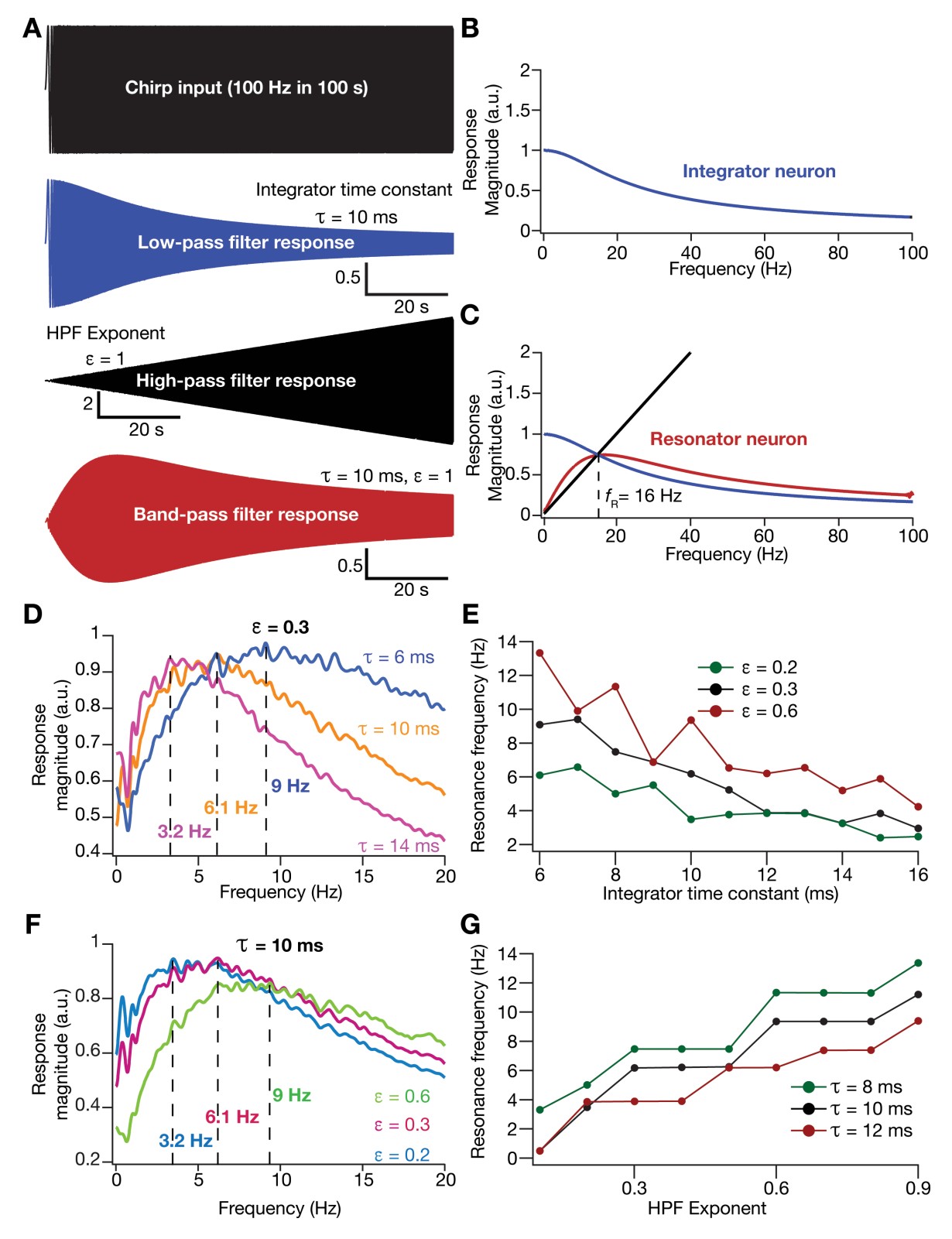

**Figure 5.** Incorporation of an additional high-pass filter into neuronal dynamics introduces resonance in individual rate-based neurons. (**A**) Responses of neurons with low-pass (integrator; blue), high-pass (black) and band-pass (resonator; red) filtering structures to a chirp stimulus (top). *Equations (7–8)* were employed for computing these responses. (**B**) Response magnitude of an integrator neuron (low-pass filter) as a function of input frequency, derived from response to the chirp stimulus. (**C**) Response magnitude of a resonator neuron (band-pass filter; red) as a function of input frequency,

*Figure 5 continued*

derived from response to the chirp stimulus, shown to emerge as a combination of low- (blue) and high-pass (black) filters. $f_R$ represents resonance frequency. The response magnitudes in (B, C) were derived from respective color-coded traces shown in (A). (D–E) Tuning resonance frequency by altering the low-pass filter characteristics. Response magnitudes of three different resonating neurons with identical HPF exponent ($\varepsilon$ = 0.3), but with different integrator time constants ($\tau$), plotted as functions of frequency (D). Resonance frequency can be tuned by adjusting $\tau$ for different fixed values of $\varepsilon$, with an increase in $\tau$ yielding a reduction in $f_R$ (E). (F–G) Tuning resonance frequency by altering the high-pass filter characteristics. Response magnitudes of three different resonating neurons with identical $\tau$ (=10 ms), but with different values for $\varepsilon$, plotted as functions of frequency (F). Resonance frequency can be tuned by adjusting $\varepsilon$ for different fixed values of $\tau$, with an increase in $\varepsilon$ yielding an increase in $f_R$ (G).

## Homogeneous CAN models constructed with phenomenological resonator neurons exhibited stable grid-patterned neural activity

How does the expression of intrinsic resonance in individual neurons of CAN models alter their grid-patterned neural activity? How do grid patterns respond to changes in resonance frequency, realized either by altering $\tau$ or $\varepsilon$? To address these, we replaced all neurons in the homogeneous CAN model with theta-frequency resonators and presented the network with the same virtual trajectory (*Figure 2E*). We found that CAN models with resonators were able to reliably and robustly produce grid-patterned neural activity, which were qualitatively (*Figure 6A,B*) and quantitatively (*Figure 6C–F*, *Figure 6—figure supplement 1*) similar to patterns produced by networks of integrator neurons across different values of $\tau$. Importantly, increase in $\tau$ markedly increased spacing between grid fields (*Figure 6D*) and their average size (*Figure 6E*), consequently reducing the number of grid fields within the arena (*Figure 6F*). It should be noted that the reduction in grid score with increased $\tau$ (*Figure 6C*) is merely a reflection of the reduction in the number of grid fields within the arena, and not indicative of loss of grid-patterned activity. Although the average firing rates were tuned to be similar across the resonator and the integrator networks (*Figure 6—figure supplement 1A*), the peak-firing rate in the resonator network was significantly higher (*Figure 6—figure supplement 1B*). In addition, for both resonator and integrator networks, consistent with increases in grid-field size and spacing, there were marked increases in information rate (*Figure 6—figure supplement 1C*) and reductions in sparsity (*Figure 6—figure supplement 1D*), with increase in $\tau$.

Within our model framework, it is important to emphasize that although increasing $\tau$ reduced $f_R$ in resonator neurons (*Figures 5E* and *6B*), the changes in grid spacing and size are not a result of change in $f_R$, but a consequence of altered $\tau$. This inference follows from the observation that altering $\tau$ has qualitatively and quantitatively similar outcomes on grid spacing and size in both integrator and resonator networks (*Figure 6*). Further confirmation for the absence of a direct role for $f_R$ in regulating grid spacing or size (within our modeling framework) came from the invariance of grid spacing and size to change in $\varepsilon$, which altered $f_R$ (*Figure 7*). Specifically, when we fixed $\tau$, and altered $\varepsilon$ across resonator neurons in the homogeneous CAN model, $f_R$ of individual neurons changed (*Figure 7A*), but did not markedly change grid field patterns (*Figure 7A*), grid score, average grid spacing, mean size, number, or sparsity of grid fields (*Figure 7B*). However, increasing $\varepsilon$ decreased grid-field sizes, the average and the peak firing rates, consequently reducing the information rate in the activity pattern (*Figure 7B*).

Together, these results demonstrated that homogeneous CAN models with phenomenological resonators reliably and robustly produce grid-patterned neural activity. In these models, the LPF regulated the size and spacing of the grid fields and the HPF governed the magnitude of activity and the suppression of low-frequency inputs.

## Phenomenological resonators stabilized the emergence of grid-patterned activity in heterogeneous CAN models

Having incorporated intrinsic resonance into the CAN model, we were now equipped to directly test our hypothesis that the expression of intrinsic neuronal resonance could stabilize grid-like activity patterns in heterogeneous networks. We incorporated heterogeneities, retaining the same forms/degrees of heterogeneities (*Table 1*; *Figure 2A–D*) and the same virtual trajectory (*Figure 2E*), in a CAN model endowed with resonators to obtain the spatial activity maps for individual neurons (*Figure 8A*). Strikingly, we observed that the presence of resonating neurons in the CAN model stabilized grid-patterned activity in individual neurons, despite the introduction of the highest degrees of all forms of biological heterogeneities (*Figure 8A*). Importantly, outputs produced by the

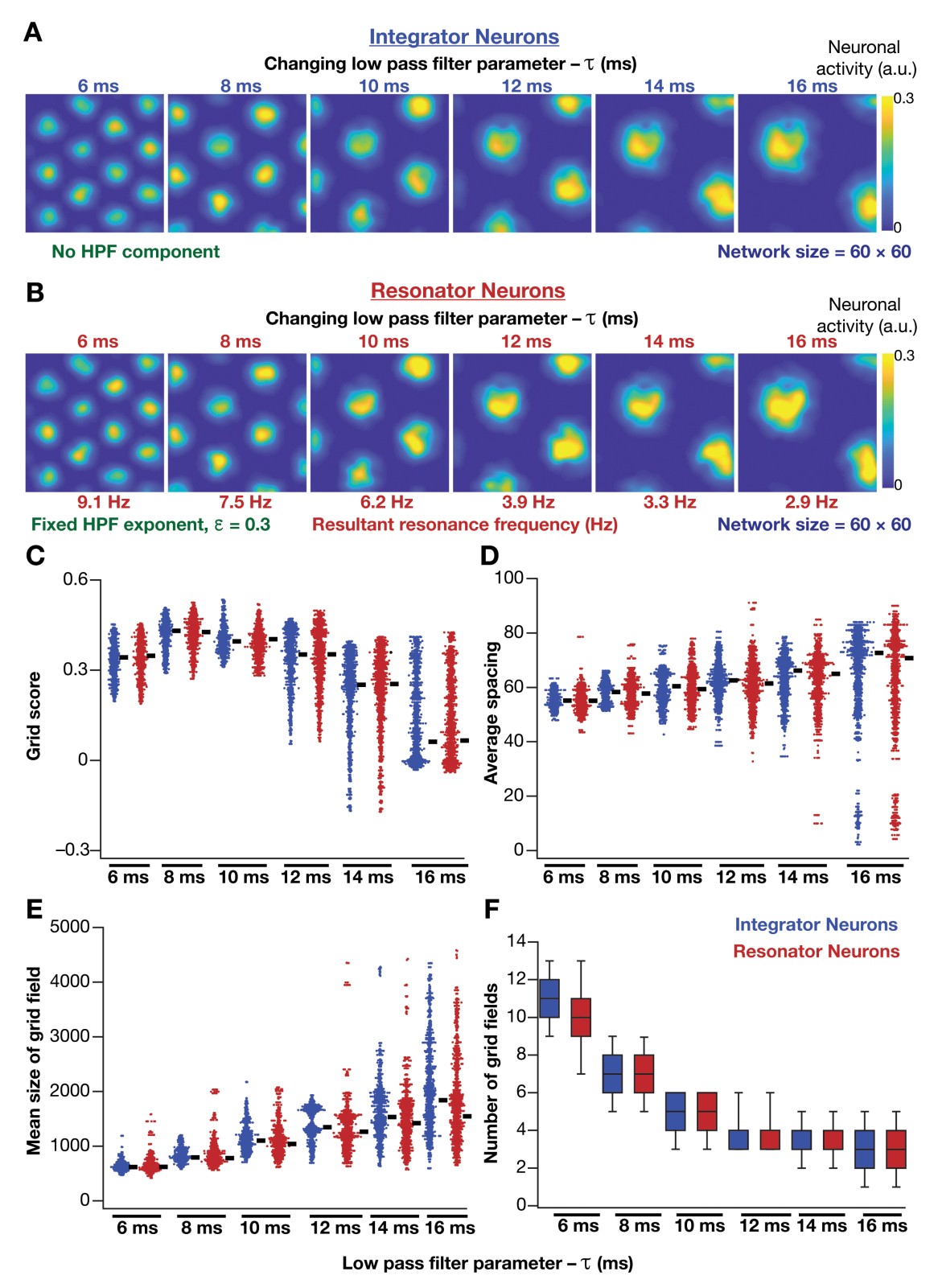

**Figure 6.** Impact of neuronal resonance, introduced by altering low-pass filter characteristics, on grid-cell activity in a homogeneous CAN model. (A) Example rate maps of grid-cell activity from a homogeneous CAN model with integrator neurons modeled with different values for integration time constants ($\tau$). (B) Example rate maps of grid-cell activity from a homogeneous CAN model with resonator neurons modeled with different $\tau$ values. (C–

*Figure 6 continued on next page*

*Figure 6 continued*

F) Grid score (C), average spacing (D), mean size (E), and number (F) of grid fields in the arena for all neurons (n = 3600) in homogeneous CAN models with integrator (blue) or resonator (red) neurons, modeled with different τ values. The HPF exponent ε was set to 0.3 for all resonator neuronal models. The online version of this article includes the following figure supplement(s) for figure 6:

**Figure supplement 1.** Impact of neuronal resonance (phenomenological model), introduced by altering low-pass filter characteristics, on grid-cell characteristics in a homogeneous CAN model.

homogeneous CAN model manifested spatially precise circular grid fields, exhibiting regular triangular firing patterns across trials (e.g., *Figure 1A*), which constituted forms of precision that are not observed in electrophysiologically obtained grid-field patterns (*Fyhn et al., 2004*; *Hafting et al., 2005*). However, that with the incorporation of biological heterogeneities in resonator neuron networks, the imprecise shapes and stable patterns of grid-patterned firing (e.g., *Figure 8A*) tend closer to those of biologically observed grid cells. Quantitatively, we computed grid-cell measurements across all the 3600 neurons in the network and found that all quantitative measurements (*Figure 8B–I*, *Figure 8—figure supplement 1*), including the grid score (*Figure 8B*) and information rate

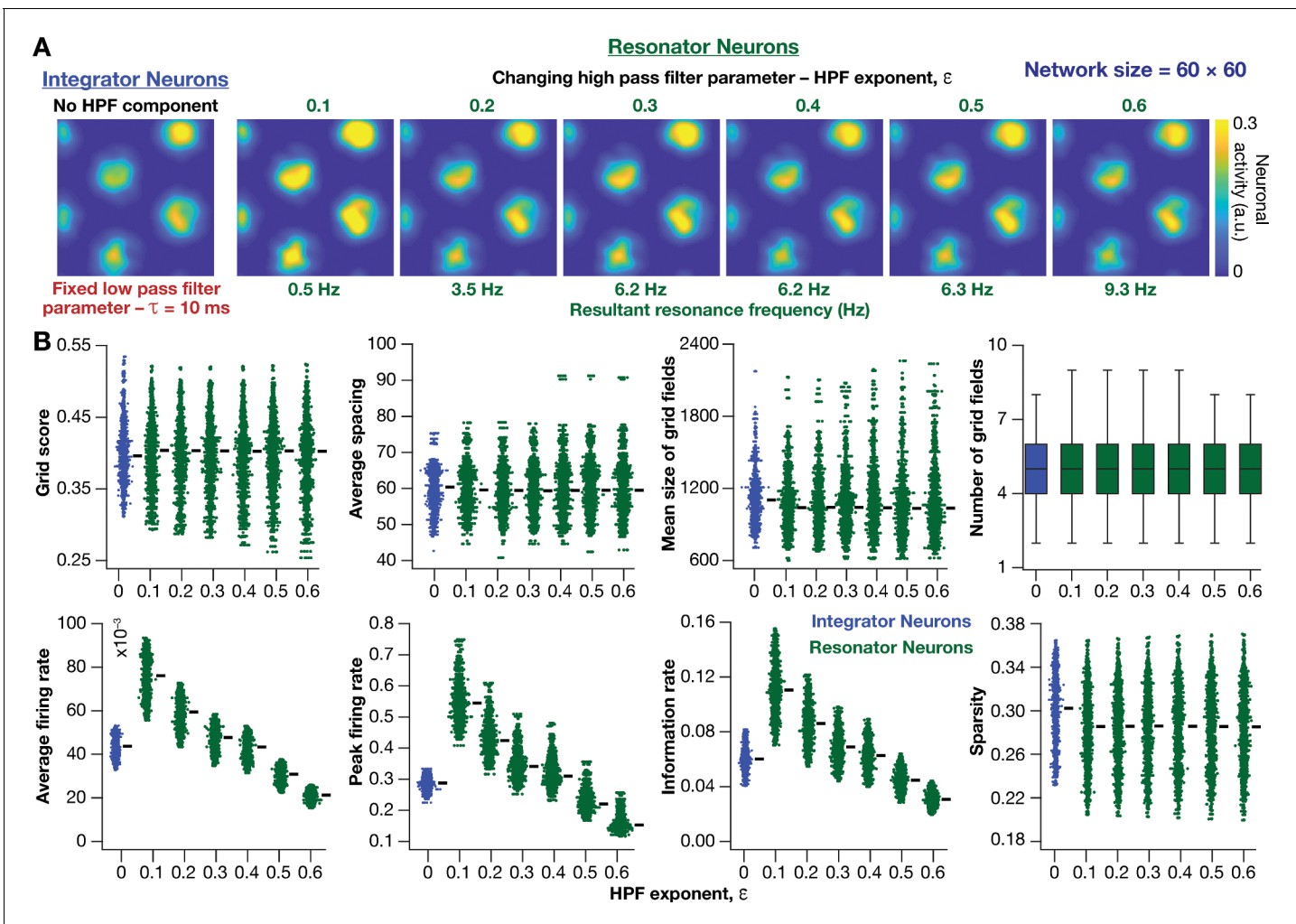

**Figure 7.** Impact of neuronal resonance, introduced by altering high-pass filter characteristics, on grid-cell activity in a homogeneous CAN model. (**A**) Example rate maps of grid-cell activity from a homogeneous CAN model with integrator neurons (*Column 1*) or resonator neurons (*Columns 2–6*) modeled with different values of the HPF exponent (ε). (**B**) Comparing 8 metrics of grid-cell activity for all the neurons (n = 3600) in CAN models with integrator (blue) or resonator (green) neurons. CAN models with resonator neurons were simulated for different ε values. τ = 10 ms for all networks depicted in this figure.

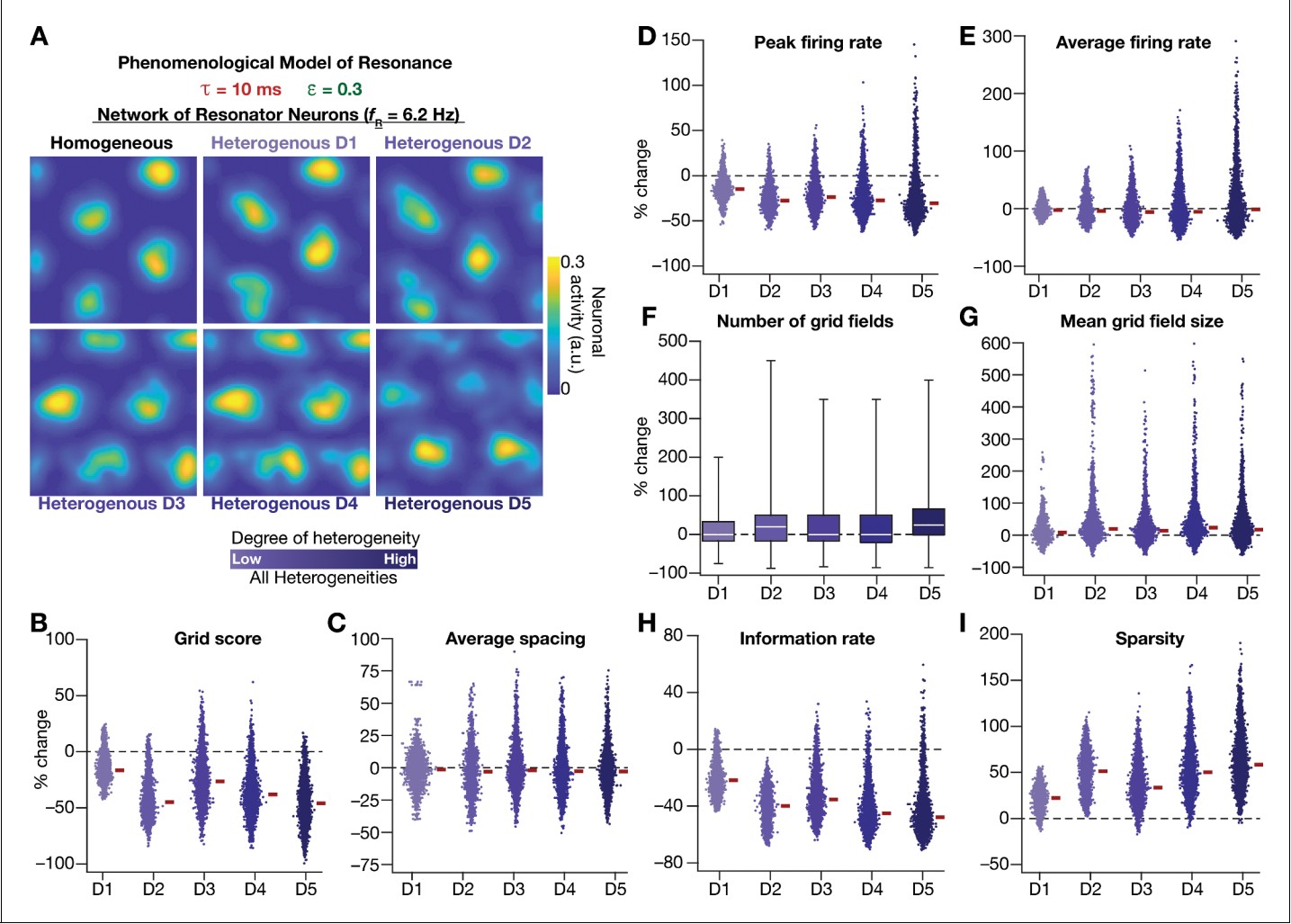

**Figure 8.** Neuronal resonance stabilizes grid-cell activity in heterogeneous CAN models. (A) Example rate maps of grid-cell activity in homogeneous (*Top left*) and heterogeneous CAN models, endowed with resonating neurons, across different degrees of heterogeneities. (B–I) Percentage changes in grid score (B), average spacing (C), peak firing rate (D), average firing rate (E), number (F), mean size (G), information rate (H), and sparsity (I) of grid field for all neurons (n = 3600) in the heterogeneous CAN model, plotted for 5 degrees of heterogeneities (D1–D5), compared with respective neurons in the homogeneous resonator network. All three forms of heterogeneities were incorporated together into the network. $\varepsilon = 0.3$ and $\tau = 10$ ms for all networks depicted in this figure.

The online version of this article includes the following figure supplement(s) for figure 8:

**Figure supplement 1.** Quantification of the grid-cell activity in presence of different forms of network heterogeneities in the CAN model with phenomenological resonator neurons.

**Figure supplement 2.** Neuronal resonance (phenomenological) stabilizes grid-cell firing in heterogeneous CAN models.

(*Figure 8H*), were remarkably robust to the introduction of heterogeneities (cf. *Figures 2–3* for CAN models with integrator neurons).

To ensure that the robust emergence of stable grid-patterned activity in heterogeneous CAN models with resonator neurons was not an artifact of the specific network under consideration, we repeated our analyses with two additional sets of integrator-resonator comparisons (*Figure 8—figure supplement 2*). In these networks, we employed a baseline $\tau$ of either 14 ms (*Figure 8—figure supplement 2A*) or 8 ms (*Figure 8—figure supplement 2B*) instead of the 10 ms value employed in the default network (*Figures 2* and *8*). A pairwise comparison of all grid-cell measurements between networks with integrator vs. resonator neurons demonstrated that the resonator networks were resilient to the introduction of heterogeneities compared to integrator networks (*Figure 8—figure supplement 2*).

Together, these results demonstrated that phenomenologically incorporating intrinsic resonance into neurons of the CAN model imparts resilience to the network, facilitating the robust emergence of stable grid-cell activity patterns, despite the expression of biological heterogeneities in neuronal and synaptic properties.

## Mechanistic model of neuronal intrinsic resonance: incorporating a slow activity-dependent negative feedback loop

In the previous sections, we had phenomenologically introduced intrinsic resonance in rate-based neurons, by incorporating an artificial high-pass filter to the low-pass filtering property of an integrator neuron. Physiologically, however, intrinsic resonance results from the presence of resonating conductances in the neural system, which introduce a slow activity-dependent negative feedback into the neuron. To elaborate, there are two requirements for any conductance to act as a resonating conductance (*Hutcheon and Yarom, 2000*). First, the current mediated by the conductance should *actively suppress* any changes in membrane voltage of neurons. In resonating conductances, this is implemented by the voltage-dependent gating properties of the ion channel mediating the resonating conductance whereby any change in membrane voltage (de)activates the channel in such a way that the channel current suppresses the change in membrane voltage. This constitutes an activity-dependent negative feedback loop. The second requirement for a resonating conductance is that it has to be *slower* compared to the membrane time constant. This requirement ensures that cutoff frequency of the lowpass filter (associated with the membrane time constant) is greater than that of the high-pass filter (mediated by the resonating conductance). Together these two requirements *mechanistically* translate to a slow activity-dependent negative feedback loop, a standard network motif in biological systems for yielding damped oscillations and resonance (*Alon, 2019*). In intrinsically resonating neurons, resonance is achieved because the resonating conductance actively suppresses low-frequency signals through this slow negative feedback loop, in conjunction with the suppression of higher frequency signals by the low-pass filter mediated by the membrane time constant. Resonating conductances do not suppress high frequencies because they are slow and would not activate/deactivate with faster high-frequency signals (*Hutcheon and Yarom, 2000*; *Narayanan and Johnston, 2008*).

In the single-neuron integrator model employed in this study, the LPF is implemented by the integration time constant $\tau$. Inspired by the mechanisms behind the physiological emergence of neuronal intrinsic resonance, we *mechanistically* incorporated intrinsic resonance into this integrator neuronal model by introducing a slow negative feedback to the single neuron dynamics (*Figure 9A*). The dynamics of the coupled evolution of neuronal activity ($S$) and the feedback state variable ($m$) are provided in *Equations (11–13)*. A sigmoidal feedback kernel ($m_\infty$) regulated the activity dependence, and the feedback time constant ($\tau_m$) determined the kinetics of the negative feedback loop (*Equations 11 and 12*). The response of this single-neuronal model, endowed with the slow-negative feedback loop, to a pulse input manifested typical sag associated with the expression of intrinsic resonance (*Hutcheon and Yarom, 2000*), owing to the slow evolution of the feedback state variable $m$ (*Figure 9B,C*). To directly test whether the introduction of the slow negative feedback loop yielded a resonating neuronal model, we assessed single-neuron output to a chirp stimulus (*Figure 9D*) and found the manifestation of intrinsic resonance when the feedback loop was activated (*Figure 9E*). As this resonator neuron model involving a slow negative feedback loop was derived from the mechanistic origins of intrinsic neuronal resonance, we refer to this as the *mechanistic resonator model*.

Resonance frequency in the mechanistic resonator model was tunable by altering the parameters associated with the feedback loop. Specifically, increasing the half-maximal activity of the feedback kernel ($S_{1/2}$) reduced resonance frequency within the tested range of values (*Figure 9F*), whereas increasing the slope of the feedback kernel ($k$) enhanced $f_R$ initially, but reduced $f_R$ with further increases in $k$ (*Figure 9G*). Furthermore, enhancing the strength of the feedback ($g$) increased $f_R$, and is reminiscent of increased $f_R$ with increase in resonating conductance density (*Narayanan and Johnston, 2008*), the parameter that defines the strength of the negative feedback loop in intrinsic resonating neurons. An important requirement for the expression of resonance through the mechanistic route that we chose is that the feedback time constant ($\tau_m$) has to be slower than the integration time constant ($\tau$ = 10 ms). To test this in our model, we altered $\tau_m$ of the feedback loop while maintaining all other values to be constant (*Figure 9I*) and found that resonance did not manifest

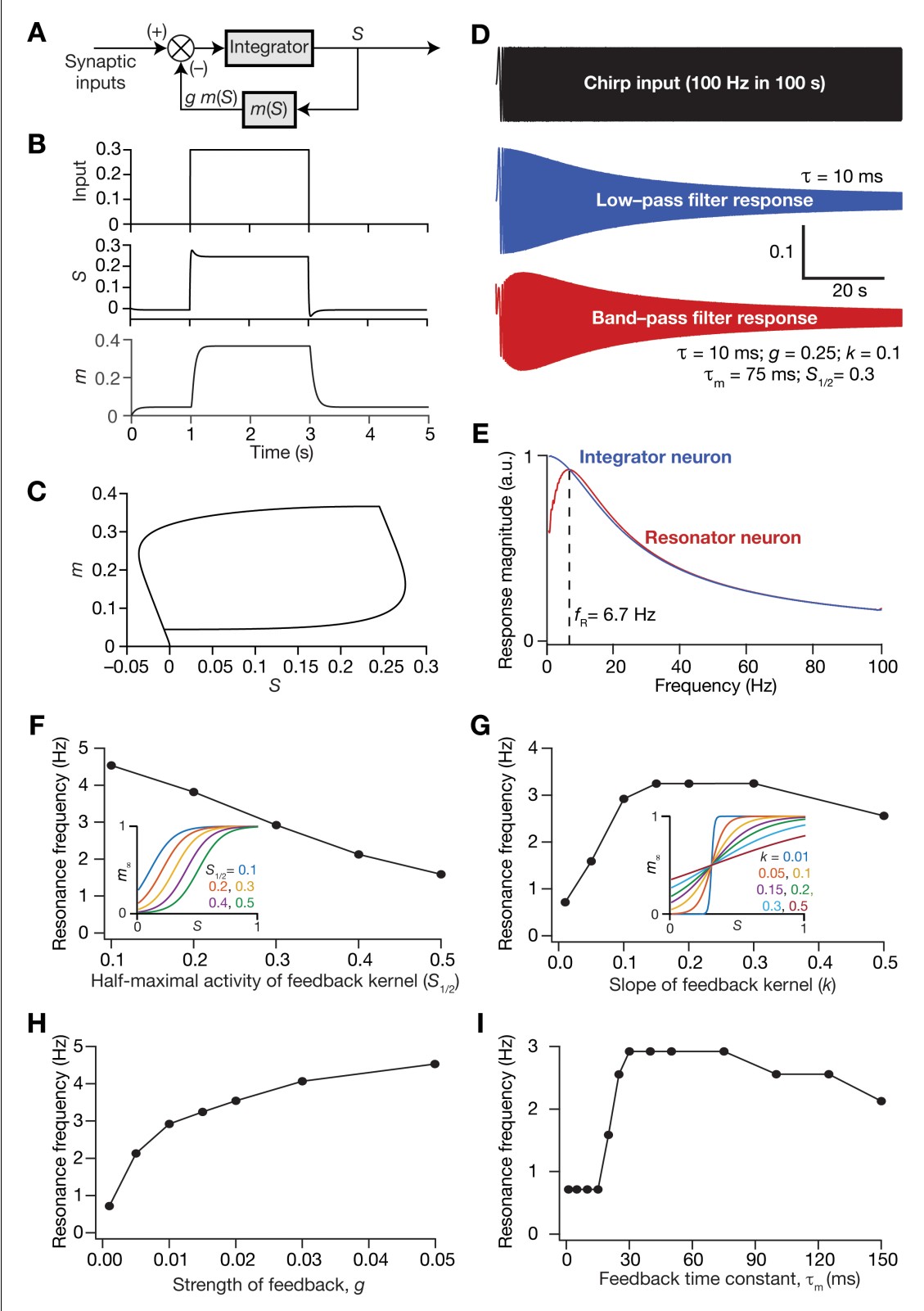

**Figure 9.** Incorporation of a slow negative feedback loop into single-neuron dynamics introduces tunable resonance in rate-based neuronal models. (A) A mechanistic model of intrinsic resonance in individual neurons using a slow negative feedback loop. (B) Temporal evolution of the output ($S$) of an individual neuron and the state variable related to the negative feedback ($m$) in response for square pulse. (C) Phase-plane representation of the dynamics depicted in (B). (D) Responses of neurons with low-pass (integrator; blue) and band-pass (resonator; red) filtering structures to a chirp stimulus

*Figure 9 continued on next page*

*Figure 9 continued*

(top). The resonator was implemented through the introduction of a slow negative feedback loop (A). *Equations (10–12)* were employed for computing these responses. (E) Response magnitude of an integrator neuron (low-pass filter, blue) and resonator neuron (band-pass filter, red) as functions of input frequency, derived from their respective responses to the chirp stimulus. $f_R$ represents resonance frequency. The response magnitudes in (E) was derived from respective color-coded traces shown in (D). (F–I) Tuning resonance frequency by altering the parameters of the slow negative feedback loop. Resonance frequency, obtained from an individual resonator neuron responding to a chirp stimulus, is plotted as functions of half maximal activity of the feedback kernel, $S_{1/2}$ (F), slope of the feedback kernel, $k$ (G), strength of negative feedback, $g$ (H), and feedback time constant, $\tau_m$ (I). The insets in (F) and (G) depict the impact of altering $S_{1/2}$ and $k$ on the feedback kernel ($m_\infty$), respectively.

with low values of $\tau_m$. With increase in $\tau_m$, there was an initial rapid increase in $f_R$, which then reduced upon further increase in $\tau_m$ (*Figure 9I*). It may be noted that low values of $\tau_m$ translate to a fast negative feedback loop, thus emphasizing the need for a slow negative feedback loop for the expression of resonance. These observations are analogous to the dependence of resonance frequency in intrinsically resonating neurons on the activation time constant of the resonating conductance (*Narayanan and Johnston, 2008*). Together, we have proposed a tunable *mechanistic* model for intrinsic resonance in rate-based neurons through the incorporation of a slow negative feedback loop to the neuronal dynamics.

## Mechanistic resonators stabilized the emergence of grid-patterned activity in heterogeneous CAN models

How was grid-patterned neural activity in a homogeneous CAN model affected by the expression of intrinsic neuronal resonance through a mechanistic framework? To address this, we replaced all the integrator neurons in the homogeneous CAN model (*Figure 2E*; homogeneous network) with mechanistic theta-frequency resonator neurons, while maintaining identical connectivity patterns and velocity dependent inputs. We observed that neurons within this homogeneous CAN model built with mechanistic resonators were able to produce distinct grid-patterned neural activity (*Figure 10A*). Sensitivity analyses showed that grid-field size and spacing were affected by the parameters associated with the slow negative feedback loop, consequently affecting the grid score (*Figure 10B*) and other measurements associated with the grid-patterned activity (*Figure 10—figure supplements 1–2*). Within the tested ranges, the strength of feedback ($g$) had limited effect on grid-patterned neural activity, whereas the slope ($k$) and half-maximal activity ($S_{1/2}$) of the feedback kernel had strong impact on grid-field size and spacing (*Figure 10*). Although the grid fields were well-defined across all values of $k$ and $S_{1/2}$, the grid scores were lower for some cases (e.g., $k$=0.5; $S_{1/2}$=0.1 in *Figure 10B*) due to the smaller number of grid fields within the arena (*Figure 10A*). Importantly, the feedback time constant ($\tau_m$) had very little impact on grid pattern neural activity (*Figure 10A*) and on all quantified measurements (*Figure 10B*, *Figure 10—figure supplements 1–2*). Together, these analyses demonstrated that a homogenous CAN model built with mechanistic resonators yielded stable and robust grid pattern neural activity whose grid field size and spacing can be tuned by parameters of slow negative feedback that governed resonance.

Next, we systematically incorporated the different degrees of heterogeneities into the CAN model with mechanistic resonator neurons to test if resonance could stabilize grid-patterned activity in the heterogeneous CAN models. We employed 5 degrees of all heterogeneities with the same virtual trajectory (*Figure 2E*) in a CAN model endowed with mechanistic resonators to compute the spatial activity maps for individual neurons (*Figure 11A*). As with the network built with phenomenological resonators (*Figure 8*), we observed stable and robust grid-like spatial maps emerge even with highest degree of heterogeneities in the CAN model with mechanistic resonator neurons (*Figure 11*). Furthermore, compared to homogeneous model, the grid pattern activity obtained by the heterogeneous CAN model closely mimicked the grid-cell activity, endowed with imprecise grid field shapes observed in neurons in MEC region of the brain (*Figure 11*). Quantitatively, the grid score (*Figure 11B*), average spacing (*Figure 11C*), and mean grid field size (*Figure 11G*) were remarkably robust to the introduction of heterogeneities. However, the average (*Figure 11E*) and peak firing (*Figure 11D*) reduced with increase in degree of heterogeneities (*Figure 11D*) and consequently affected measurements that were dependent on firing rate (i.e., information rate and sparsity; *Figure 11H–I*). Together, these results demonstrate that intrinsic neuronal resonance, introduced

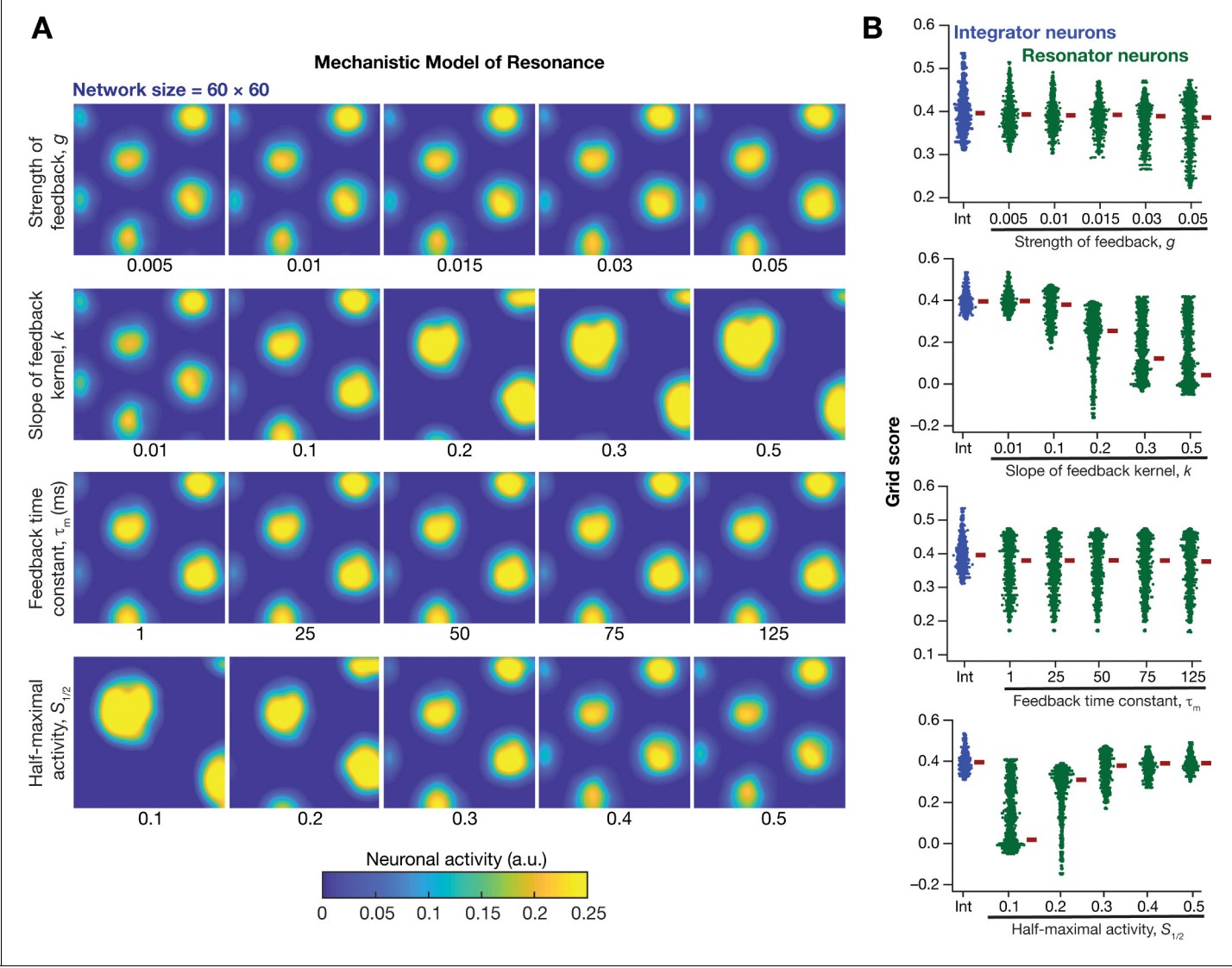

**Figure 10.** Impact of neuronal resonance, introduced by a slow negative feedback loop, on grid-cell activity in a homogeneous CAN model. (**A**) Example rate maps of grid-cell activity from a homogeneous CAN model for different values of the feedback strength ($g$) slope of the feedback kernel ($k$), feedback time constant ($\tau_m$), and half maximal activity of the feedback kernel, $S_{1/2}$. (**B**) Grid scores for all the neurons in the homogeneous CAN model for different values of $g$, $k$, $\tau_m$, and $S_{1/2}$ in resonator neurons (green). Grid scores for homogeneous CAN models with integrator neurons (without the negative feedback loop) are also shown (blue). Note that although pattern neural activity is observed across all networks, the grid score is lower in some cases because of the large size and lower numbers of grid fields within the arena with those parametric configurations.

The online version of this article includes the following figure supplement(s) for figure 10:

**Figure supplement 1.** Impact of intrinsic neuronal resonance, introduced by adding a negative feedback loop in the neuronal dynamics, on grid-cell characteristics in a homogeneous CAN model.

**Figure supplement 2.** Impact of intrinsic neuronal resonance, introduced by adding a negative feedback loop in the neuronal dynamics, on grid-cell characteristics in a homogeneous CAN model.

either through phenomenological (*Figure 8*) or through mechanistic (*Figure 11*) single-neuron models, yielded stable and robust grid-like pattern in heterogeneous CAN models.

The grid-cell activity on a 2D plane represents a spatially repetitive periodic pattern. Consequently, a phase plane constructed from activity along the diagonal of the 2D plane on one axis (*Figure 11—figure supplement 1A*), and the spatial derivative of this activity on the other should provide a visual representation of the impact of heterogeneities and resonance on this periodic pattern of activity. To visualize the specific changes in the periodic orbits obtained with homogeneous

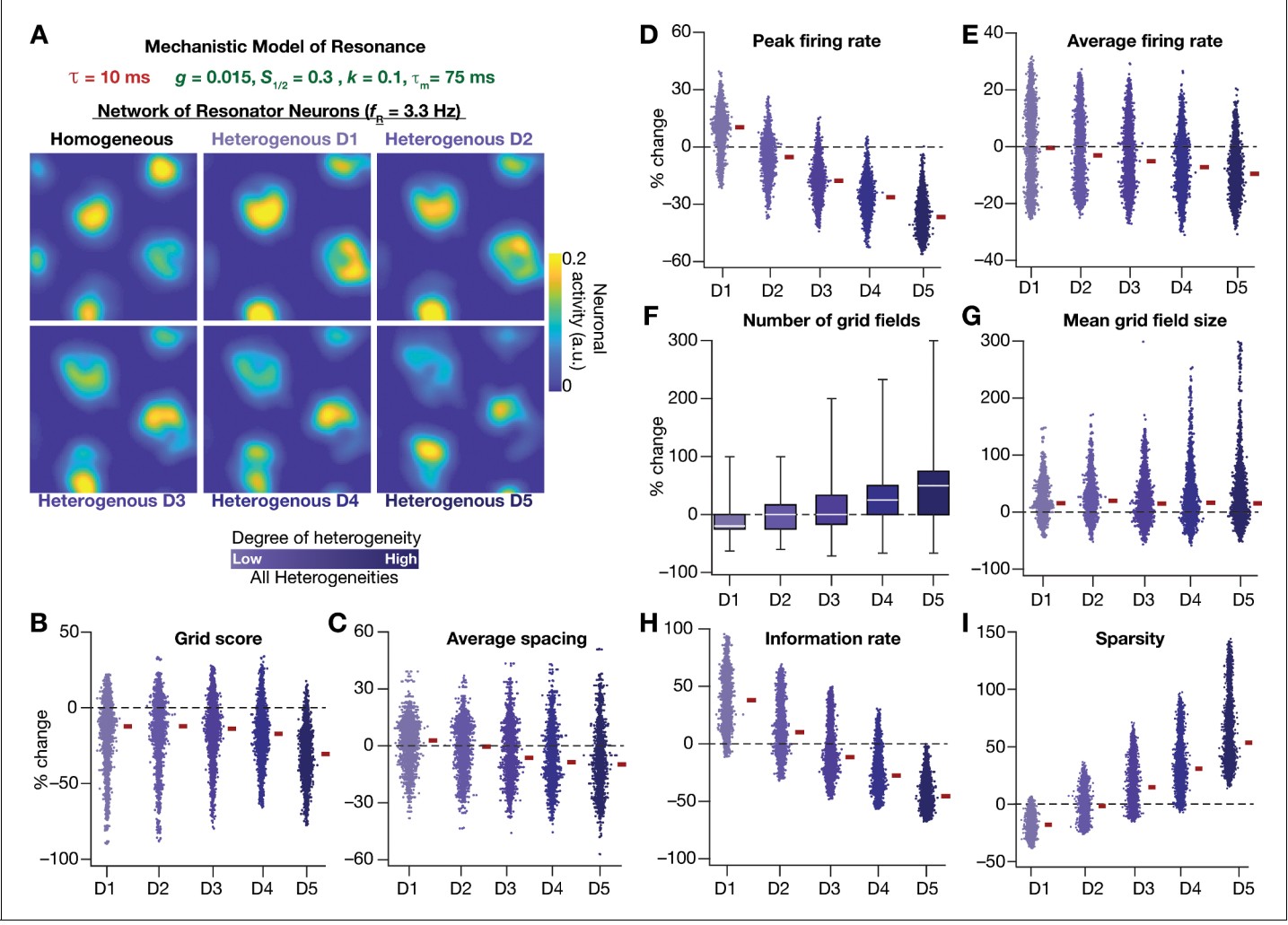

**Figure 11.** Resonating neurons, achieved through a slow negative feedback loop, stabilizes grid-cell activity in heterogeneous CAN models. (A) Example rate maps of grid-cell activity in homogeneous (top left) and heterogeneous CAN models, endowed with resonating neurons, across different degrees of heterogeneities. (B–I) Percentage changes in grid score (B), average spacing (C), peak firing rate (D), average firing rate (E), number (F), mean size (G), information rate (H), and sparsity (I) of grid field for all neurons (n = 3600) in the heterogeneous CAN model, plotted for 5 degrees of heterogeneities (D1–D5). The percentage changes are computed with reference to respective neurons in the homogeneous resonator network. All three forms of heterogeneities were incorporated together into these networks.

The online version of this article includes the following figure supplement(s) for figure 11:

**Figure supplement 1.** Phase plane analysis of spatial profiles provided visualizations of the disruption of grid-cell activity in heterogeneous integrator networks and the relative robustness of heterogeneous resonator networks.

and heterogeneous networks constructed with integrators or resonators, we plotted the diagonal activity profiles of five randomly picked neurons using the phase plane representation (*Figure 11— figure supplement 1B*). This visual representation confirmed that homogenous CAN models built of integrators or resonators yielded stable closed orbit trajectories, representing similarly robust grid-patterned periodic activity (*Figure 11—figure supplement 1B*). The disparate sizes of different orbits are merely a reflection of the intersection of the diagonal with a given grid field. Specifically, if the grid fields were considered to be ideal circles, the orbital size is the largest if diagonal passes through the diameter, with orbital size reducing for any other chord. Upon introduction of heterogeneities, the phase-plane plots of diagonal activity profiles from the heterogeneous integrator network lacked closed orbits (*Figure 11—figure supplement 1B*). This is consistent with drastic reductions in grid score and information rate for these heterogeneous network models (*Figures 2– 3*). In striking contrast, the phase-plane plots from the heterogeneous resonator network manifested

closed orbits even with the highest degree of heterogeneities introduced, irrespective of whether resonance was introduced through a phenomenological or a mechanistic model (*Figure 11—figure supplement 1B*). Although these phase-plane trajectories were noisy compared to those from the homogeneous resonator network, the presence of closed orbits indicated the manifestation of spatial periodicity in these activity patterns (*Figure 11—figure supplement 1B*). These observations visually demonstrated that resonator neurons stabilized the heterogeneous network and maintained spatial periodicity in grid-cell activity, irrespective of whether resonance was introduced through a phenomenological or a mechanistic model.

## The slow kinetics of the negative feedback loop in mechanistic resonators is a critical requirement for stabilizing heterogeneous CAN models

Our rationale behind the introduction of intrinsic resonance into the neuron was that it would suppress the low-frequency perturbations introduced by biological heterogeneities (*Figure 4*). Consequently, our hypothesis is that the *targeted* suppression of low-frequency components resulted in the stabilization of the CAN model. Two lines of evidence for this hypothesis were that the suppression of low-frequency components through phenomenological (*Figure 8*) or through mechanistic (*Figure 11*) resonators resulted in stabilization of the heterogeneous CAN models. An advantage of recruiting a mechanistic model for introducing resonance is that sensitivity analyses on its parameters could provide valuable mechanistic insights about how such stabilization is achieved. With reference to specific hypothesis, the ability to tune resonance by altering the feedback time constant $\tau_m$ (*Figure 9I*), without altering the feedback kernel or the feedback strength, provides an efficient route to understand the mechanistic origins of the stabilization. Specifically, the value of $\tau_m$ (default value = 75 ms) governs the slow kinetics of the feedback loop and is the source for the *targeted* suppression of low-frequency components. Reducing $\tau_m$ would imply a faster negative feedback loop, thereby suppressing even higher-frequency components. As a further test of our hypothesis on the role of suppressing low-frequency components in stabilizing the CAN network, we asked if mechanistic resonators with lower values of $\tau_m$ would be able to stabilize heterogeneous CAN models (*Figure 12*). If fast negative feedback loops (i.e., low values of $\tau_m$) were sufficient to stabilize heterogeneous CAN models, that would counter our hypothesis on the requirement of targeted suppression of low-frequency components. To the contrary, we found that heterogeneous CAN networks with neurons endowed with fast negative feedback loops were incapable of stabilizing the grid-cell network (*Figure 12*). With progressive increase in $\tau_m$, the grid-patterned firing stabilized even for high degrees of heterogeneities (*Figure 12B,C* for low values of $\tau_m$; *Figure 11B* for $\tau_m$ = 75 ms), thus providing a direct additional line of evidence supporting our hypothesis on the need for targeted suppression of low-frequency components in stabilizing the network. We noted that the impact of altering $\tau_m$ was specifically related to stabilizing heterogeneous networks by suppressing heterogeneities-driven low-frequency perturbations because altering $\tau_m$ did not alter grid-patterned activity in homogeneous resonator networks (*Figure 9A,B*). Together, our results provide multiple lines of evidence that the slow kinetics of the negative feedback loop in single-neuron dynamics (*Figure 9A*) mediates targeted suppression of low-frequency signals (*Figure 9D,E*), thereby yielding intrinsic neuronal resonance (*Figure 9I*) and stabilizing grid-patterned activity in heterogeneous CAN models (*Figures 11* and *12)*.

## Resonating neurons suppressed the impact of biological heterogeneities on low-frequency components of neural activity

As detailed above, our hypothesis on a potential role for intrinsic neuronal resonance in stabilizing grid-patterned firing in heterogeneous CAN models was centered on the ability of resonators in *specifically* suppressing low-frequency components. Although our analyses provided evidence for stabilization of grid-patterned firing with phenomenological (*Figure 8*) or mechanistic (*Figure 11*) resonators, did resonance specifically suppress the impact of biological heterogeneities of low-frequency components of neural activity? To assess this, we performed frequency-dependent analyses of neural activity in CAN models with phenomenological or mechanistic resonators (*Figure 13*, *Figure 13—figure supplements 1–3*) and compared them with integrator-based CAN models (*Figure 4*). First, we found that the variance in the deviations of neural activity in heterogeneous

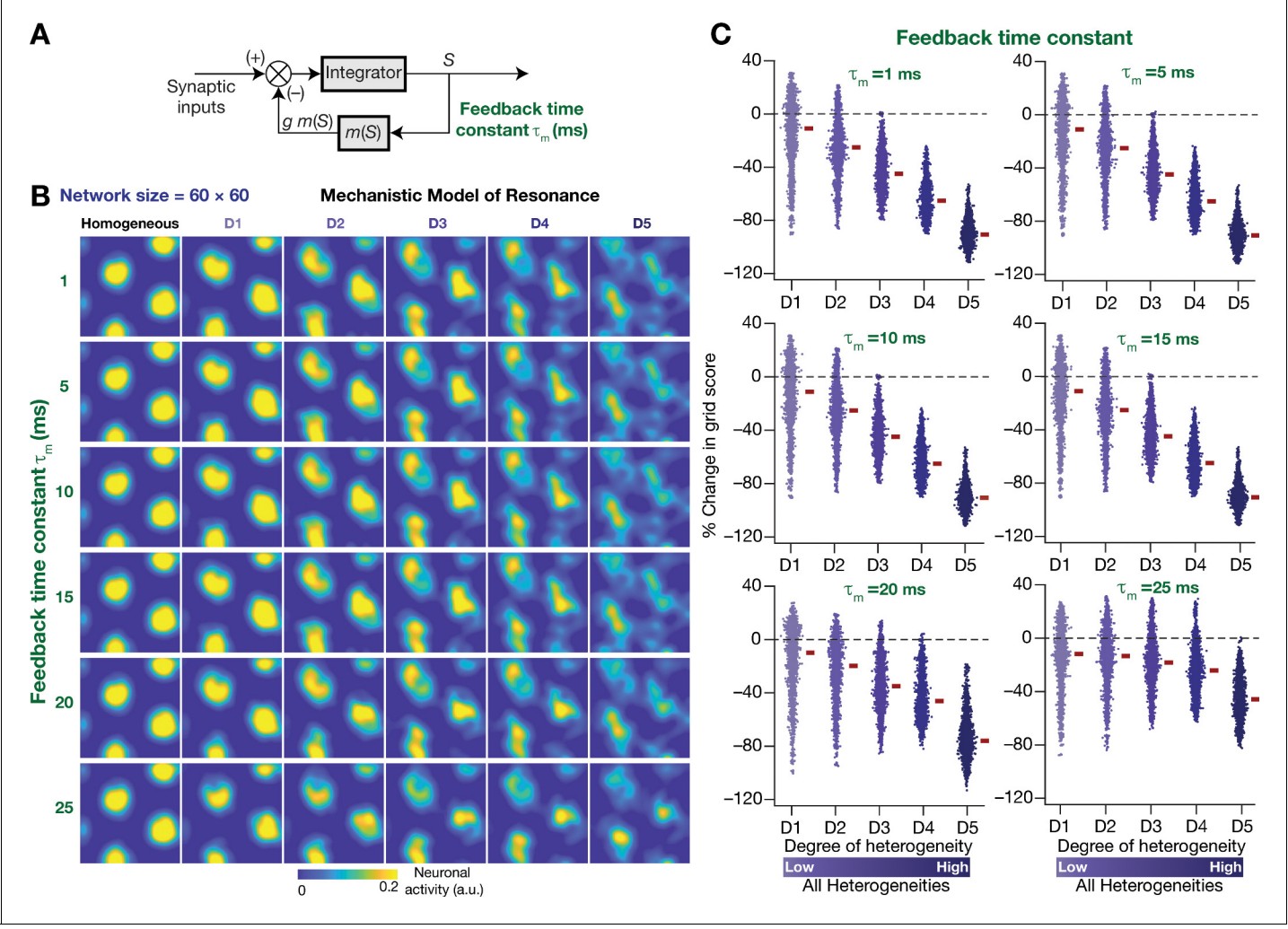

**Figure 12.** The slow kinetics of the negative feedback loop is a critical requirement for stabilizing heterogeneous CAN models. (**A**) A mechanistic model of intrinsic resonance in individual neurons using a slow negative feedback loop, with the feedback time constant ($\tau_m$) defining the slow kinetics. (**B**) Example rate maps of grid-cell activity in homogeneous (top left) and heterogeneous CAN models, endowed with neurons built with different values of $\tau_m$, across different degrees of heterogeneities. (**C**) Percentage changes in grid score for all neurons (n = 3600) in the heterogeneous CAN model, endowed with neurons built with different values of $\tau_m$, plotted for 5 degrees of heterogeneities (D1–D5). The percentage changes are computed with reference to respective neurons in the homogeneous resonator network. All three forms of heterogeneities were incorporated together into these networks.

networks with reference to the respective homogeneous models were considerably lower with resonator neurons (both phenomenological and mechanistic models), compared to their integrator counterparts (*Figure 13A,B*, *Figure 13D,E*, *Figure 13—figure supplements 1–3*). Second, comparing the *relative* power of neural activity across different octaves, we found that network with resonators suppressed lower frequencies (predominantly the 0–2 Hz band) and enhanced power in the range of neuronal resonance frequencies, when compared with their integrator counterparts (*Figure 13C*, *Figure 13F*, *Figure 13—figure supplements 1–3*). This relative suppression of low-frequency power accompanied by the relative enhancement of high-frequency power was observed across all networks with resonator, either homogeneous (*Figure 13—figure supplements 1–2*) or heterogeneous with distinct forms of heterogeneities (*Figure 13—figure supplement 3G–I*). Importantly, given the slow activity-dependent negative feedback loop involved in mechanistic resonators, the low-frequency suppression was found to be extremely effective across all degrees of heterogeneities (*Figure 13D,E*) with minimal increases of power in high-frequency bands (*Figure 13F*) compared to their phenomenological counterparts (*Figure 13A–C*). The phenomenological resonators were built

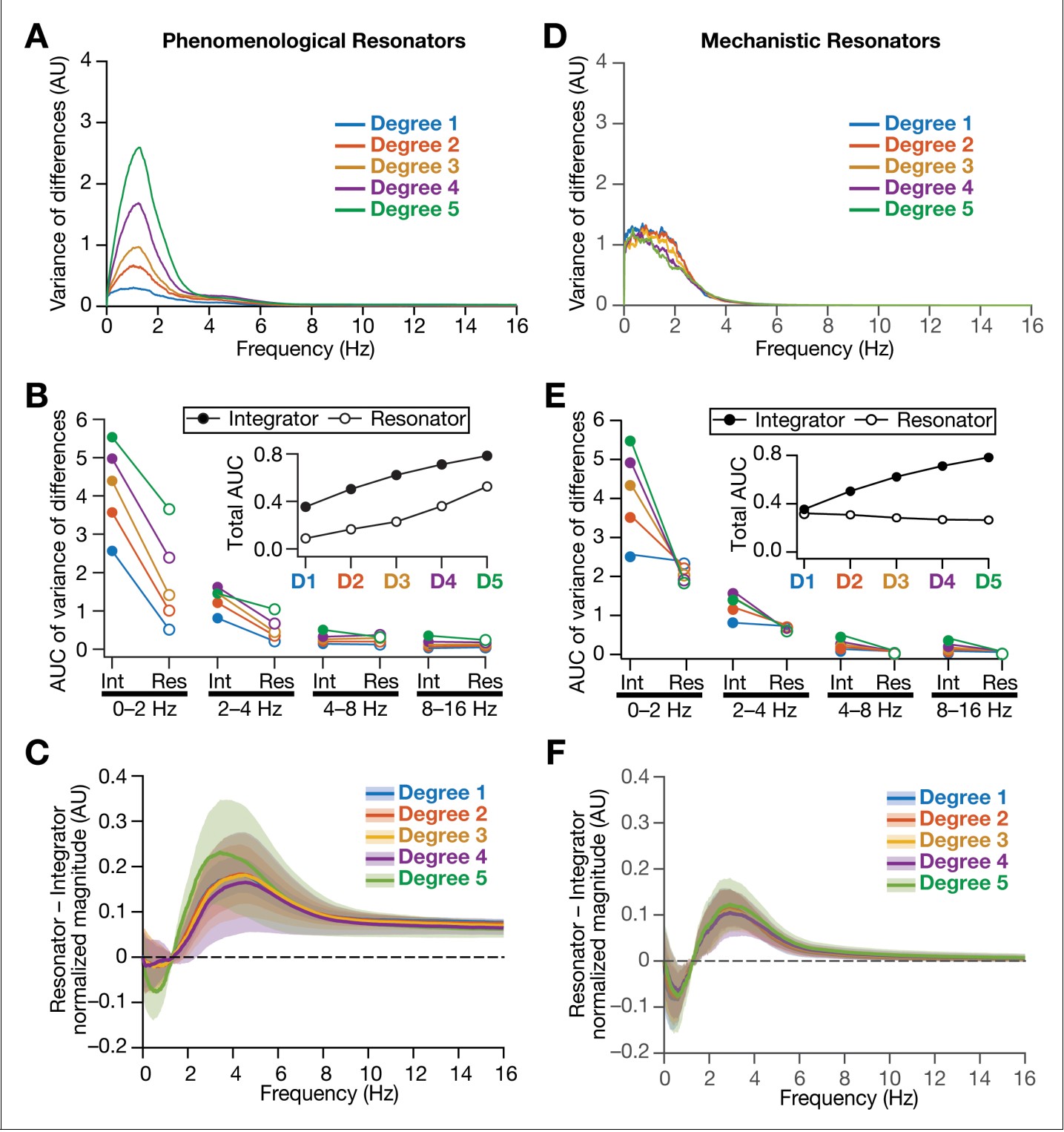

**Figure 13.** Intrinsically resonating neurons suppressed heterogeneity-induced variability in low-frequency perturbations caused by the incorporation of biological heterogeneities. (A) Normalized variance of the differences between the magnitude spectra of temporal activity in neurons of homogeneous vs. heterogeneous networks, across different degrees of all three forms of heterogeneities expressed together, plotted as a function of frequency. (B) Area under the curve (AUC) of the normalized variance plots shown in *Figure 4H* (for the integrator network) and (A) (for the phenomenological resonator network) showing the variance to be lower in resonator networks compared to integrator networks. The inset shows the total AUC across all frequencies for the integrator vs. the resonator networks as a function of the degree of heterogeneities. (C) Difference between the normalized magnitude spectra of neural temporal activity patterns for integrator and resonator neurons in CAN models. Solid lines depict the mean and shaded

*Figure 13 continued on next page*

*Figure 13 continued*

area depicts the standard deviations, across all 3600 neurons. The resonator networks in (**A–C**) were built with phenomenological resonators. (**D–F**) same as (**A–C**) but for the mechanistic model of intrinsic resonance. All heterogeneities were simultaneously expressed for all the analyses presented in this figure.

The online version of this article includes the following figure supplement(s) for figure 13:

**Figure supplement 1.** Intrinsically resonating neurons (phenomenological) suppressed low-frequency components and enhanced frequency components around resonance frequency in homogeneous CAN models.

**Figure supplement 2.** Intrinsically resonating neurons (mechanistic) suppressed low-frequency components and enhanced frequency components around resonance frequency in homogeneous CAN models.

**Figure supplement 3.** Intrinsically resonating neurons (phenomenological) suppressed heterogeneity-induced variability in low-frequency perturbations caused by different forms of biological heterogeneities.

with a simple HPF that is not endowed with activity-dependent filtering capabilities. In addition, the derivative-based implementation of the phenomenological resonator model yielded spurious high-frequency power, which was averted with the slow activity-dependent negative feedback loop incorporated into the mechanistic resonator model (*Figure 13D–F*).

Together, our results demonstrated that biological heterogeneities predominantly altered low-frequency components of neural activity, and provided strong quantitative lines of evidence that intrinsic neuronal resonance plays a stabilizing role in heterogeneous networks by targeted suppression of low-frequency inputs, thereby counteracting the disruptive impact of biological heterogeneities on low-frequency components.

## Discussion

The principal conclusion of this study is that intrinsic neuronal resonance stabilizes heterogeneous 2D CAN models, by suppressing heterogeneity-induced perturbations in low-frequency components of neural activity. Our analyses provided the following lines of evidence in support of this conclusion:

1. Neural-circuit heterogeneities destabilized grid-patterned activity generation in a 2D CAN model (*Figures 2* and *3*).
2. Neural-circuit heterogeneities predominantly introduced perturbations in the low-frequency components of neural activity (*Figure 4*).
3. Targeted suppression of low-frequency components through phenomenological (*Figure 5C*) or through mechanistic (*Figure 9D*) resonators resulted in stabilization of the heterogeneous CAN models (*Figures 8* and *11*). Thus, stabilization was achieved irrespective of the means employed to suppress low-frequency components: an activity-independent suppression of low frequencies (*Figure 5*) or an activity-dependent slow negative feedback loop (*Figure 9*).
4. Changing the feedback time constant $\tau_m$ in mechanistic resonators, *without* changes to neural gain or the feedback strength allowed us to control the specific range of frequencies that would be suppressed. Our analyses showed that a slow negative feedback loop, which results in targeted suppression of low-frequency components, was essential in stabilizing grid-patterned activity (*Figure 12*). As the slow negative feedback loop and the resultant suppression of low frequencies *mediates* intrinsic neuronal resonance, these analyses provide important lines of evidence for the role of targeted suppression of low frequencies in stabilizing grid-patterned activity.
5. We demonstrate that the incorporation of phenomenological (*Figure 13A–C*) or mechanistic (*Figure 13D–F*) resonators specifically suppressed lower frequencies of activity in the 2D CAN model.

### A physiological role for intrinsic neuronal resonance in stabilizing heterogeneous neural networks

Intrinsic neuronal resonance is effectuated by the expression of resonating conductances, which are mediated by restorative channels endowed with activation time constants slower than the neuronal membrane time constant (*Hutcheon and Yarom, 2000*). The gating properties and the kinetics of these ion channels allow them to suppress low-frequency activity with little to no impact on higher frequencies, yielding resonance in conjunction with the LPF governed by the membrane time constant (*Hutcheon and Yarom, 2000*). This intrinsic form of neuronal resonance, mediated by ion

channels that are intrinsic to the neural structure, is dependent on membrane voltage (*Narayanan and Johnston, 2007*; *Narayanan and Johnston, 2008*; *Hu et al., 2009*), somato-dendritic recording location (*Narayanan and Johnston, 2007*; *Narayanan and Johnston, 2008*; *Hu et al., 2009*), neuronal location along the dorso-ventral axis (*Giocomo et al., 2007*; *Garden et al., 2008*) and is regulated by activity-dependent plasticity (*Brager and Johnston, 2007*; *Narayanan and Johnston, 2007*). Resonating conductances have been implicated in providing frequency selectivity to specific range of frequencies under sub- and supra-threshold conditions (*Hutcheon and Yarom, 2000*; *Narayanan and Johnston, 2007*; *Hu et al., 2009*; *Das and Narayanan, 2017*; *Das et al., 2017*), in mediating membrane potential oscillations (*Fransén et al., 2004*; *Mittal and Narayanan, 2018*), in mediating coincidence detection through alteration to the class of neural excitability (*Das and Narayanan, 2017*; *Das et al., 2017*), in regulating spatio-temporal summation of synaptic inputs (*Garden et al., 2008*), in introducing phase leads in specific ranges of frequencies regulating temporal relationships of neural responses to oscillatory inputs (*Narayanan and Johnston, 2008*), in regulating local-field potentials and associated spike phases (*Sinha and Narayanan, 2015*; *Ness et al., 2018*), in mediating activity homeostasis through regulation of neural excitability (*Brager and Johnston, 2007*; *Narayanan and Johnston, 2007*; *Honnuraiah and Narayanan, 2013*), in altering synaptic plasticity profiles (*Nolan et al., 2007*; *Narayanan and Johnston, 2010*), and in regulating grid-cell scales (*Giocomo et al., 2007*; *Giocomo et al., 2011b*; *Giocomo et al., 2011a*; *Pastoll et al., 2012*).

Our analyses provide an important additional role for intrinsic resonance in stabilizing heterogeneous neural network, through the suppression of heterogeneity-induced perturbation in low-frequency components. The $1/f$ characteristics associated with neural activity implies that the power in lower frequencies is higher (*Buzsaki, 2006*), and our analyses show that the incorporation of biological heterogeneities into networks disrupt their functional outcomes by introducing perturbations predominantly in the lower frequencies. We demonstrate that resonating conductances, through their ability to suppress low-frequency activity, are ideally placed to suppress such low-frequency perturbations thereby stabilizing activity patterns in the face of biological heterogeneities. As biological heterogeneities are ubiquitous, we postulate intrinsic neuronal resonance as a powerful mechanism that lends stability across different heterogeneous networks, through suppression of low-frequency perturbations introduced by the heterogeneities. A corollary to this postulate is that the specific resonance frequency of a given neural structure is a reflection of the need to suppress frequencies where the perturbations are introduced by specific forms and degree of biological heterogeneities expressed in the network where the structure resides.

Within this framework, the dorso-ventral gradient in resonance frequencies in entorhinal stellate cells could be a reflection of the impact of specific forms of heterogeneities expressed along the dorso-ventral axis on specific frequency ranges, with resonance frequency being higher in regions where a larger suppression of low-frequency components is essential. Future electrophysiological and computational studies could specifically test this hypothesis by quantifying the different heterogeneities in these sub-regions, assessing their frequency-dependent impact on neural activity in individual neurons and their networks. More broadly, future studies could directly measure the impact of perturbing resonating conductances on network stability and low-frequency activity in different brain regions to test our predictions on the relationship among biological heterogeneities, intrinsic resonance, and network stability.

## Slow negative feedback: stability, noise suppression, and robustness

Our analyses demonstrated the efficacy of a slow negative feedback loop in stabilizing grid-patterned activity in CAN models (*Figures 11–12*). From a broader perspective, negative feedback is a standard motif for establishing stabilization and robustness of dynamical systems spanning control engineering (*Nyquist, 1932*; *Black, 1934*; *Bode, 1945*; *Bechhoefer, 2005*; *Åström and Murray, 2008*) and biological networks (*Barkai and Leibler, 1997*; *Weng et al., 1999*; *Hutcheon and Yarom, 2000*; *Bhalla et al., 2002*; *Milo et al., 2002*; *Shen-Orr et al., 2002*; *Tyson et al., 2003*; *Alon, 2007*; *Turrigiano, 2007*; *Novák and Tyson, 2008*; *Tyson and Novák, 2010*). From the perspective of biological systems, the impact of negative feedback on stability, robustness, and homeostasis spans multiple scales, from single molecules dynamics to effective functioning of entire ecosystem. Examples of the stabilizing roles of negative feedback at the organ level include baroreflex in blood pressure regulation (*Sved, 2009*), body temperature, blood glucose level, endocrine

hormone secretion (*Modell et al., 2015*), and erythropoiesis (*Koulnis et al., 2014*). Cells are equipped with distinct receptors that can sense changes in temperature, pH, damage to cellular workhorse proteins or DNA, internal state of cells and accumulation of products (*Tyson and Novák, 2010*). Using functional motifs consisting combinations of positive and negative feedback loops often imparts stability to the biochemical reactions and signaling networks comprising these receptors, thus maintaining homeostasis (*Alon, 2007*). Specific examples for negative feedback in biochemical reactions and signaling networks include protein synthesis (*Goodwin, 1966*), mitotic oscillators (*Goldbeter, 1991*), MAPK signaling pathways (*Kholodenko, 2000*; *Bhalla et al., 2002*), cAMP production (*Martiel and Goldbeter, 1987*), adaptation in bacterial chemotaxis (*Knox et al., 1986*; *Barkai and Leibler, 1997*; *Spiro et al., 1997*; *Yi et al., 2000*; *Levchenko and Iglesias, 2002*), and M/G1 module (G1/S and G2/M phases) of the cell cycle control system (*Rupes, 2002*).

Importantly, with specific relevance to our hypothesis, negative feedback has been shown to reduce the effects of noise and enhance system stability with reference to internal and external perturbations because of the suppressive nature of this motif (*Savageau, 1974*; *Becskei and Serrano, 2000*; *Thattai and van Oudenaarden, 2001*; *Austin et al., 2006*; *Dublanche et al., 2006*; *Raj and van Oudenaarden, 2008*; *Lestas et al., 2010*; *Cheong et al., 2011*; *Voliotis et al., 2014*). In addition, negative feedback alleviates bottlenecks on information transfer (*Cheong et al., 2011*), and has been proposed as a mechanism for stable alignment of dose-response in the signaling system for mating pheromones in yeast, by adjusting the dose-response of downstream systems to match the dose-response of upper-level systems and simultaneously reducing the amplification of stochastic noise in the system (*Yu et al., 2008*).

At the cellular scale, the dynamics of action potential, which is a fundamental unit of information transfer in nervous system, is critically dependent on the negative feedback by delayed-rectifier potassium channels (*Hodgkin and Huxley, 1952c*; *Hodgkin and Huxley, 1952b*; *Hodgkin and Huxley, 1952a*). Resonating conductances (or phenomenological inductances) mediate slow negative feedback in sustaining sub-threshold membrane potential oscillations along with amplifying conductances that mediate a fast positive feedback loop (*Cole and Baker, 1941*; *Mauro, 1961*; *Cole, 1968*; *Mauro et al., 1970*; *Hutcheon and Yarom, 2000*; *Narayanan and Johnston, 2008*). A similar combination of positive and negative feedback loop involving synaptic connectivity has been suggested to modulate the neuronal oscillation frequency during sensory processing in neuronal circuit model of layers 2 and 3 of sensory cortex (*Lee et al., 2018*). Furthermore, positive and negative feedback signals play critical roles in the emergence of neuronal polarity (*Takano et al., 2019*). Activity-dependent negative feedback mechanisms control the density of ion channels and receptors based on the levels of neural activity, resulting in homeostatic activity regulation (*Bienenstock et al., 1982*; *Turrigiano et al., 1994*; *Turrigiano et al., 1995*; *Turrigiano et al., 1998*; *Desai et al., 1999*; *Turrigiano, 2007*; *O'Leary et al., 2010*; *O'Leary and Wyllie, 2011*; *Honnuraiah and Narayanan, 2013*; *O'Leary et al., 2014*; *Srikanth and Narayanan, 2015*). Microglia have been shown to stabilize neuronal activity through a negative feedback loop that is dependent on extracellular ATP concentration, which is dependent on neural activity (*Badimon et al., 2020*). Models for neuronal networks have successfully utilized the negative feedback loop for achieving transfer of decorrelated inputs of olfactory information in the paleocortex (*Ambros-Ingerson et al., 1990*), in the reduction of redundancy in information transfer in the visual pathway (*Pece, 1992*), in a model of the LGN inhibited by the V1 to achieve specificity (*Murphy and Sillito, 1987*), and in a model of cerebellum based on feedback motifs (*D'Angelo et al., 2016*).

Finally, an important distinction between the phenomenological and the mechanistic resonators is that the former is activity independent, whereas the latter is activity dependent in terms of their ability to suppress low-frequency signals. This distinction explains the differences between how these two resonators act on homogeneous and heterogeneous CAN networks, especially in terms of suppressing low-frequency power (*Figure 13*, *Figure 13—figure supplements 1–3*). Together, the incorporation of resonance through a negative feedback loop allowed us to link our analyses to the well-established role of network motifs involving negative feedback loops in inducing stability and suppressing external/internal noise in engineering and biological systems. We envisage intrinsic neuronal resonance as a cellular-scale activity-dependent negative feedback mechanism, a specific instance of a well-established network motif that effectuates stability and suppresses perturbations across different networks.

## Future directions and considerations in model interpretation

Our analyses here employed a rate-based CAN model for the generation of grid-patterned activity. Rate-based models are inherently limited in their ability to assess temporal relationships between spike timings, which are important from the temporal coding perspective where spike timings with reference to extracellular oscillations have been shown to carry spatial information within grid fields (*Hafting et al., 2008*). Additionally, the CAN model is one of the several theoretical and computational frameworks that have been proposed for the emergence of grid-cell activity patterns, and there are lines of experimental evidence that support aspects of these distinct models (*Kropff and Treves, 2008*; *Burak and Fiete, 2009*; *Burgess and O'Keefe, 2011*; *Giocomo et al., 2011a*; *Navratilova et al., 2012*; *Couey et al., 2013*; *Domnisoru et al., 2013*; *Schmidt-Hieber and Häusser, 2013*; *Yoon et al., 2013*; *Schmidt-Hieber et al., 2017*; *Urdapilleta et al., 2017*; *Stella et al., 2020*; *Tukker et al., 2021*). Within some of these frameworks, resonating conductances have been postulated to play specific roles, distinct from the one proposed in our study, in the emergence of grid-patterned activity and the regulation of their properties (*Giocomo et al., 2007*; *Giocomo et al., 2011b*; *Giocomo et al., 2011a*; *Pastoll et al., 2012*). Together, the use of the rate-based CAN model has limitations in terms of assessing temporal relationships between oscillations and spike timings, and in deciphering the other potential roles of resonating conductances in grid-patterned firing. However, models that employ other theoretical frameworks do not explicitly incorporate the several heterogeneities in afferent, intrinsic, and synaptic properties of biological networks, including those in the conductance and gating properties of resonating conductances. Future studies should therefore explore the role of resonating conductances in stabilizing conductance-based grid-cell networks that are endowed with all forms of biological heterogeneities. Such conductance-based analyses should also systematically assess the impact of resonating conductances, their kinetics, and gating properties (including associated heterogeneities) in regulating temporal relationships of spike timings with theta-frequency oscillations spanning the different theoretical frameworks.

A further direction for future studies could be the use of morphologically realistic conductance-based model neurons, which would enable the incorporation of the distinct ion channels and receptors distributed across the somato-dendritic arborization. Such models could assess the role of interactions among several somato-dendritic conductances, especially with resonating conductances, in regulating grid-patterned activity generation (*Burgess and O'Keefe, 2011*; *Giocomo et al., 2011a*). In addition, computations performed by such a morphologically realistic conductance-based neuron are more complex than the simplification of neural computation as an integrator or a resonator (*Schmidt-Hieber et al., 2017*). For instance, owing to the differential distribution of ionic conductances, different parts of the neurons could exhibit integrator- or resonator-like characteristics, with interactions among different compartments yielding the final outcome (*Das and Narayanan, 2017*; *Das et al., 2017*). The conclusions of our study emphasizing the importance of biological heterogeneities and resonating conductances in grid-cell models underline the need for heterogeneous, morphologically realistic conductance-based network models to systematically compare different theoretical frameworks for grid-cell emergence. Future studies should endeavor to build such complex heterogeneous networks, endowed with synaptic and channel noise, in systematically assessing the role of heterogeneities and specific ion-channel conductances in the emergence of grid-patterned neural activity across different theoretical frameworks.

In summary, our analyses demonstrated that incorporation of different forms of biological heterogeneities disrupted network functions through perturbations that were predominantly in the lower frequency components. We showed that intrinsic neuronal resonance, a mechanism that suppressed low-frequency activity, stabilized network function. As biological heterogeneities are ubiquitous and as the dominance of low-frequency perturbations is pervasive across biological networks (*Hausdorff and Peng, 1996*; *Gilden, 2001*; *Gisiger, 2001*; *Ward, 2001*; *Buzsaki, 2006*), we postulate that mechanisms that suppress low-frequency components could be a generalized route to stabilize heterogeneous biological networks.

## Materials and methods

### Development of a virtual trajectory to reduce computational cost

We developed the following algorithm to yield faster virtual trajectories in either a circular (2 m diameter) or a square (2 m × 2 m) arena:

1. The starting location of the virtual animal was set at the center of the arena ($x_0 = 1$, $y_0 = 1$).
2. At each time step (=1 ms), two random numbers were picked, one denoting distance from a uniform distribution ($d_t \in [0, 0.004]$) and another yielding the angle of movement from another uniform distribution ($A_t \in [-\pi/36, \pi/36]$). The angle of movement was restricted to within $\pi/36$ on either side to yield smooth movements within the spatial arena. The new coordinate of the animal was then updated as:

$$x_t = x_{t-1} + d_t \sin(A_t) \tag{1}$$

$$y_t = y_{t-1} + d_t \cos(A_t) \tag{2}$$

   If the new location ($x_t$, $y_t$) fell outside the arena, the $d_t$ and $A_t$ are repicked until ($x_t$, $y_t$) were inside the bounds of the arena.
3. To enable sharp turns near the boundaries of the arena, the $A_t$ random variable was picked from a uniform distribution of ($A_t \in [0, 2\pi]$) instead of uniform distribution of ($A_t \in [-\pi/36, \pi/36]$) if either $x_{t-1}$ or $y_{t-1}$ was close to arena boundaries. This enhanced range for $A_t$ closed to the boundaries ensured that there was enough deflection in the trajectory to mimic sharp turns in animal runs in open arena boundaries.

The limited range of $A_t \in [-\pi/36, \pi/36]$ in step two ensured that the head direction and velocity inputs to the neurons in the CAN model were not changing drastically at every time step of the simulation run, thereby stabilizing spatial activity. We found the virtual trajectories yielded by this algorithm to closely mimic animal runs in an open arena, with better control over simulation periods and better computational efficiency in terms of covering the entire arena within shorter time duration.

### Structure and dynamics of the continuous attractor network model with integrator neurons

Each neuron in the network has a preferred direction $\theta_i$ (assumed to receive input from specific head direction cells), which can take its value to be among 0, $\pi/2$, $\pi$, and $3\pi/2$, respectively, depicting east, north, west, and south. The network was divided into local blocks of four cells representing each of these four directions and local blocks were uniformly tiled to span the entire network. This organization translated to a scenario where one-fourth of the neurons in the network were endowed with inputs that had the same direction preference. Of the two sets of synaptic inputs to neurons in the network, intra-network inputs followed a Mexican-hat connectivity pattern. The recurrent weights matrix for Mexican-hat connectivity was achieved through the difference of Gaussian equation, given by *Burak and Fiete, 2009*:

$$W_{ij} = W_0 \left( x_i - x_j - l\hat{e}_{\theta j} \right) \tag{3}$$

$$W_0(\boldsymbol{x}) = a \exp(-\gamma|\boldsymbol{x}|^2) - \exp(-\beta|\boldsymbol{x}|^2) \tag{4}$$

where $W_{ij}$ represented the synaptic weight from neuron $j$ (located at $x_j$) to neuron $i$ (located at $x_i$), $\hat{e}_{\theta j}$ defined the unit vector pointing along the $\theta_j$ direction. This weight matrix was endowed with a center-shifted center-surround connectivity, and the parameter $l$ (default value: 2) defined the amount of shift along $\hat{e}_{\theta j}$. In the difference of Gaussians formulation in *Equation 6*, the parameter $a$ regulated the sign of the synaptic weights and was set to 1, defining an all-inhibitory network. The other parameters were $\gamma = 1.1 \times \beta$ with $\beta = 3/\lambda^2$, and $\lambda$ (default value: 13) defining the periodicity of the lattice (*Burak and Fiete, 2009*).

The second set of synaptic inputs to individual neurons, arriving as feed-forward inputs based on the velocity of the animal and the preferred direction of the neuron, was computed as:

$$B_i = 1 + \alpha \hat{e}_{\theta j} \cdot \mathbf{v} \tag{5}$$

where $\alpha$ denotes a gain parameter for the velocity inputs (velocity scaling factor), $\mathbf{v}$ represents the velocity vector derived from the trajectory of the virtual animal.

The dynamics of the rate-based integrator neurons, driven by these two sets of inputs was then computed as:

$$\tau \frac{dS_i}{dt} + S_i = f\left(\sum_j W_{ij}S_j + B_i\right) \tag{6}$$

where $f$ represented the neural transfer function, which was implemented as a simple rectification non-linearity, and $S_i(t)$ denoted the activity of neuron $i$ at time point $t$. The default value of the integration time constant ($\tau$) of neural response was 10 ms. CAN models were initialized with randomized values of $S_i$ ($S_i^0 \in [0, 1]$) for all neurons. For stable spontaneous pattern to emerge over the neural lattice, an initial constant feed-forward drive was provided by ensuring the velocity input was zero for the initial 100 ms period. The time step ($dt$) for numerical integration was 0.5 ms when we employed the real trajectory and 1 ms for simulations with virtual trajectories. We confirmed that the use of 1 ms time steps for virtual trajectories did not hamper accuracy of the outcomes. Activity patterns of all neurons were recorded for each time step. For visualization of the results, the spike threshold was set at 0.1 (a.u.).

## Incorporating biological heterogeneities into the CAN model

We introduced intrinsic, afferent, and synaptic forms of biological heterogeneities by independently randomizing the values of integration time constant ($\tau$), velocity scaling factor ($\alpha$), and the connectivity matrix ($W_{ij}$), respectively, across neurons in the CAN model. Specifically, in the homogeneous CAN model, these parameters were set to a default value and were identical across all neurons in the network. However, in heterogeneous networks, each neuron in the network was assigned a different value for these parameters, each picked from respective uniform distributions (*Table 1*). We progressively expanded the ranges of the respective uniform distributions to progressively increase the degree of heterogeneity (*Table 1*). We built CAN models with four different forms of heterogeneities: networks that were endowed with one of intrinsic, afferent, and synaptic forms of heterogeneities, and networks that expressed all forms together. In networks that expressed only one form of heterogeneity, the other two parameters were set identical across all neurons. In networks expressing all forms of heterogeneities, all three parameters were randomized with the span of the uniform distribution for each parameter concurrently increasing with the degree of heterogeneity (*Table 1*). We simulated different trials of a CAN model by employing different sets of initial randomization of activity values ($S_i^0$) for all the cells, while keeping all other model parameters (including the connectivity matrix, trajectory, and the specific instance of heterogeneities) unchanged (*Figure 3—figure supplement 1*).

## Introducing resonance in rate-based neurons: phenomenological model

To assess the frequency-dependent response properties, we employed a chirp stimulus $c_{100}(t)$, defined a constant-amplitude sinusoidal input with its frequency linearly increasing as a function of time, spanning 0–100 Hz in 100 s. We fed $c_{100}(t)$ as the input to a rate-based model neuron and recorded the response of the neuron:

$$\tau \frac{ds}{dt} + s = c_{100} \tag{7}$$

We computed the Fourier transform of the response $s(t)$ as $S(f)$ and employed the magnitude of $S(f)$ to evaluate the frequency-dependent response properties of the neuron. Expectedly, the integrator neuron acted as a LPF (*Figure 5A,B*).

A simple means to elicit resonance from the response of a low-pass system is to feed the output of the low-pass system to a HPF, and the interaction between these filters results in resonance (*Hutcheon and Yarom, 2000*; *Narayanan and Johnston, 2008*). We tested this employing the $c_{100}(t)$ stimulus, by using the low-pass response $s(t)$ to the chirp stimulus from *equation 7*:

$$h = s\left(\frac{ds}{dt}\right)^{\varepsilon} \tag{8}$$

Here, $h(t)$ represented the output of the resultant resonator neuron, $\varepsilon$ defined an exponent that regulates the slope of the frequency-selective response properties of the high-pass filter. When $\varepsilon = 0$, $h(t)$ trivially falls back to the low-pass response $s(t)$. The magnitude of the Fourier transform of $h(t)$, $H(f)$ manifested resonance in response to the $c_{100}(t)$ stimulus (*Figure 5A,C*). This model for achieving resonance in single neurons was referred to as a *phenomenological resonator*.

Having confirmed that the incorporation of a HPF would yield resonating dynamics, we employed this formulation to define the dynamics of a resonator neuron in the CAN model through the combination of the existing low-pass kinetics (*Equation 6*) and the high-pass kinetics. Specifically, we obtained $S_i$ for each neuron $i$ in the CAN model from *Equation 6* and redefined neural activity as the product of this $S_i$ (from *Equation 6*) and its derivative raised to an exponent:

$$S_i := RS_i \left( \frac{dS_i}{dt} \right)^\varepsilon \tag{9}$$

where $R$ was a scaling factor for matching the response of resonator neurons with integrator neurons and $\varepsilon$ defined the exponent of the high-pass filter. Together, whereas the frequency-dependent response of the integrator is controlled by integration time constant ($\tau$), that of a resonator is dependent on $\tau$ of the integrator as well as the HPF exponent $\varepsilon$ (*Figure 5D,E*).

To simulate homogeneous CAN models with resonator neurons, all integrator neurons in the standard homogeneous model (*Equations 3–6*) were replaced with resonator neurons. Intrinsic, synaptic, and afferent heterogeneities are introduced as previously described (*Table 1*) to build heterogeneous CAN models with resonator neurons. The other processes, including initialization procedure of the resonator neuron network, were identical to the integrator neuron network. In simulations where $\tau$ was changed from its base value of 10 ms, the span of uniform distributions that defined the five degrees of heterogeneities were appropriately rescaled and centered at the new value of $\tau$. For instance, when $\tau$ was set to 8 ms, the spans of the uniform distribution for the first and the fifth degrees of heterogeneity were 6.4–9.6 ms (20% of the base value on either side) and 1–16 ms (100% of the base value on either side, with an absolute cutoff at 1 ms), respectively.

## Mechanistic model for introducing intrinsic resonance in rate-based neurons

Neuronal intrinsic resonance was achieved by incorporating a slow negative feedback to the single-neuronal dynamics of rate-based neurons (*Figure 9A*). We tested the emergence of intrinsic resonance using the $c_{100}(t)$ stimulus described earlier (*Equation 7*; *Figure 9D–E*). The dynamics of mechanistic model of resonance as follows with the $c_{100}(t)$ stimulus:

$$\tau \frac{dS}{dt} = -S - g\, m(S) + c_{100} \tag{10}$$

$$\frac{dm}{dt} = \frac{m_\infty - m}{\tau_m} \tag{11}$$

Here, $S$ governed neuronal activity, $m$ defined the feedback state variable, and $g$ (default value 0.015) represented feedback strength. The slow kinetics of the negative feedback was controlled by the feedback time constant ($\tau_m$) with default value of 75 ms. In order to manifest resonance, $\tau_m > \tau$. The steady-state feedback kernel ($m_\infty$) of the negative feedback is sigmoidally dependent on the output of the neuron ($S$), with default value of the half-maximal activity ($S_{1/2}$) to be 0.3 and the slope ($k$) to be 0.1:

$$m_\infty = \left( 1 + \exp\left( \frac{S_{1/2} - S}{k} \right) \right)^{-1} \tag{12}$$

The magnitude of the Fourier transform of $S(t)$ in this system of differential equations, $S(f)$, was assessed for the expression of resonance in response to the $c_{100}(t)$ stimulus (*Figure 9D*). This model for achieving resonance in single neurons was referred to as a *mechanistic resonator*.

These resonating neurons were incorporated within the CAN framework to assess how neuronal intrinsic resonance achieved through mechanistic means affected the grid-cell firing. The synaptic

weight matrix (*Equation 4*) as well as the velocity dependence (*Equation 5*) associated with CAN model consisting of resonator neurons were identical to CAN model with integrator neurons, with the only difference in the single-neuronal dynamics:

$$\tau \frac{dS_i}{dt} = -S_i - g\,m(S_i) + f\left(\sum_j W_{ij}S_j + B_i\right) \tag{13}$$

with $m(S_i)$ evolving as per *Equation (11)*, implemented the activity-dependent slow negative feedback which is dependent on the current state $(S_i)$ of the $i^{\text{th}}$ neuron.

## Quantitative analysis of grid-cell activity

To quantitatively assess the impact of heterogeneities and the introduction of resonance into the CAN neurons, we employed standard measurements of grid-cell activity (*Fyhn et al., 2004*; *Hafting et al., 2005*).

We divided the space of the open arena into 100 × 100 pixels to compute the rate maps of grid-cell activity in the network. Activity spatial maps were constructed for each cell by taking the activity ($S_i$ for cell $i$) of the cell at each time stamp and synchronizing the index of corresponding $(x, y)$ location of the virtual animal in the open arena, for the entire duration of the simulation. Occupancy spatial maps were constructed by computing the probability ($p_m$) of finding the rat in $m^{\text{th}}$ pixel, employing the relative time spent by the animal in that pixel across the total simulation period. Spatial rate maps for each cell in the network were computed by normalizing their activity spatial maps by the occupancy spatial map for that run. Spatial rate maps were smoothened using a 2D Gaussian kernel with standard deviation ($\sigma$) of 2 pixels (e.g., *Figure 1A*, panels in the bottom row for each value of $T_{\text{run}}$). *Average firing rate* (μ) of grid cells was computed by summing the activity of all the pixels from rate map matrix and dividing this quantity by the total number of pixels in the rate map (N = 10,000). *Peak firing rate* ($\mu_{\text{max}}$) of a neuron was defined as the highest activity value observed across all pixels of its spatial rate map.

For estimation of grid fields, we first detected all local maxima in the 2D-smoothed version of the spatial rate maps of all neurons in the CAN model. Individual grid-firing fields were identified as contiguous regions around the local maxima, spatially spreading to a minimum of 20% activity relative to the activity at the location of the local maxima. The *number of grid fields* corresponding to each grid cell was calculated as the total number of local peaks detected in the spatial rate map of the cell. The *mean size of the grid fields* for a specific cell was estimated by calculating the ratio between the total number of pixels covered by all grid fields and the number of grid fields. The *average grid-field spacing* for individual grid cells was computed as the ratio of the sum of distances between the local peaks of all grid fields in the rate map to the total number of distance values.

*Grid score* was computed by assessing rotational symmetry in the spatial rate map. Specifically, for each neuron in the CAN model, the spatial autocorrelation value, $SAC_{\varphi}$, was computed between its spatial rate map and the map rotated by $\varphi°$, for different values of $\varphi$ (30, 60, 90, 120, or 150). These $SAC_{\varphi}$ values were used for calculating the grid score for the given cell as:

$$\text{GridScore} = \min(SAC_{60},\ SAC_{120}) - \max(SAC_{30},\ SAC_{90},\ SAC_{150}) \tag{14}$$

*Spatial information rate* ($I_S$) in bits per second was calculated by:

$$I_S = \sum_m p_m\,\mu_m\,\log_2\frac{\mu_m}{\mu} \tag{15}$$

where $p_m$ defined the probability of finding the rat in $m^{\text{th}}$ pixel, $\mu_m$ represented the mean firing rate of the grid cell in the $m^{\text{th}}$ pixel, and μ denoted the average firing rate of grid cell. *Sparsity* was computed as the ratio between the square mean rate and mean square rate:

$$\text{Sparsity} = \frac{\mu^2}{\sum_i p_i \mu_i^2} \tag{16}$$

## Quantitative analysis of grid-cell temporal activity in the spectral domain

To understand the impact of network heterogeneities on spectral properties of grid-cell activities under the CAN framework, we used the Fourier transform of the temporal activity of all the grid cells in a network. First, we assessed the difference in the magnitude spectra of temporal activity of the grid cells (n = 3600) in the homogeneous network compared to the corresponding grid cells in the heterogeneous networks (e.g., *Figure 4A*). Next, we normalized this difference in magnitude spectra for each grid cell with respect to the sum of their respective maximum magnitude for the homogeneous and the heterogeneous networks. Quantitatively, if $S_{het}(f)$ and $S_{homo}(f)$ defined neuronal activity in spectral domain for a neuron in a heterogeneous network and for the same neuron in the corresponding homogeneous network, respectively, then the normalized difference was computed as:

$$\Delta S_{het-homo}(f) = \frac{S_{het}(f) - S_{homo}(f)}{\max(S_{het}(f)) + \max(S_{homo}(f))} \tag{17}$$

Note that this normalization was essential to account for potential differences in the maximum values of $S_{het}(f)$ and $S_{homo}(f)$. Finally, we computed the variance of this normalized difference ($\Delta S_{het-homo}(f)$) across all the cells in the networks (e.g., *Figure 4B*).

In addition to using these normalized differences for quantifying spectral signatures of neural activity, we performed octave analysis on the magnitude spectra of the temporal activity of the grid cells to confirm the impact of heterogeneities or resonating neurons on different frequency octaves. Specifically, we computed the percentage of area under the curve (AUC) for each octave (0–2 Hz, 2–4 Hz, 4–8 Hz, and 8–16 Hz) from the magnitude spectra (*Figure 13—figure supplement 1B,C*). We performed a similar octave analysis on the variance of normalized difference for networks endowed with integrator or resonator neurons (*Figure 13B*, *Figure 13—figure supplement 3D–F*).

## Computational details

Grid-cell network simulations were performed in MATLAB 2018a (Mathworks Inc, USA) with a simulation step size of 1 ms, unless otherwise specified. All data analyses and plotting were performed using custom-written software within the IGOR Pro (Wavemetrics, USA) or MATLAB environments, and all statistical analyses were performed using the R statistical package (http://www.R-project.org/). To avoid false interpretations and to emphasize heterogeneities in simulation outcomes, the entire range of measurements are reported in figures rather than providing only the summary statistics (*Rathour and Narayanan, 2019*).

## Acknowledgements

The authors thank Dr. Poonam Mishra, Dr. Sufyan Ashhad, and the members of the cellular neurophysiology laboratory for helpful discussions and for comments on a draft of this manuscript. The authors thank Dr. Ila Fiete for helpful discussions. This work was supported by the Wellcome Trust-DBT India Alliance (Senior fellowship to RN; IA/S/16/2/502727), Human Frontier Science Program (HFSP) Organization (RN), the Department of Biotechnology through the DBT-IISc partnership program (RN), the Revati and Satya Nadham Atluri Chair Professorship (RN), and the Ministry of Human Resource Development (RN and DM).

## Additional information

### Funding

| Funder | Grant reference number | Author |
|---|---|---|
| Wellcome Trust DBT India Alliance | IA/S/16/2/502727 | Rishikesh Narayanan |
| Human Frontier Science Program | Career development award | Rishikesh Narayanan |
| Department of Biotechnology , | DBT-IISc partnership | Rishikesh Narayanan |

| | | |
|---|---|---|
| Ministry of Science and Technology | Program | |
| Indian Institute of Science | Revati and Satya Nadham Atluri Chair Professorship | Rishikesh Narayanan |
| Ministry of Human Resource Development | Scholarship funds | Divyansh Mittal |

The funders had no role in study design, data collection and interpretation, or the decision to submit the work for publication.

### Author contributions

Divyansh Mittal, Conceptualization, Software, Formal analysis, Validation, Investigation, Visualization, Methodology, Writing - original draft, Writing - review and editing; Rishikesh Narayanan, Conceptualization, Resources, Formal analysis, Supervision, Funding acquisition, Investigation, Visualization, Methodology, Writing - original draft, Project administration, Writing - review and editing

### Author ORCIDs

Divyansh Mittal https://orcid.org/0000-0003-4233-8176
Rishikesh Narayanan https://orcid.org/0000-0002-1362-4635

### Decision letter and Author response

Decision letter https://doi.org/10.7554/eLife.66804.sa1
Author response https://doi.org/10.7554/eLife.66804.sa2

## Additional files

### Supplementary files

- Source code 1. Source code for simulations reported in eLife.66804.

- Transparent reporting form

### Data availability

All data generated or analyzed during this study are included in the manuscript and supporting files. The custom-written simulations and analyses code (in MATLAB) employed for simulations are available as source code.

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
