## [Decision Letter]

**Acceptance summary:**

Grid cells in the rodent entorhinal cortex are believed to contribute to internal representations of space and other continuous quantities via periodic firing patterns. Using extensive simulations, Mittal and Narayanan show that a leading continuous attractor model of how such patterns emerge is fragile to biologically relevant heterogeneities. The authors show how this fragility is rescued by introducing intrinsic resonance in the dynamics of cells in the network. Such resonance is widely observed the entorhinal system. This work therefore shows an important potential role for single cell properties in regulating network-level computations.

**Decision letter after peer review:**

Thank you for submitting your article "Resonating neurons stabilize heterogeneous grid-cell networks" for consideration by *eLife*. Your article has been reviewed by 2 peer reviewers, and the evaluation has been overseen by a Reviewing Editor and Ronald Calabrese as the Senior Editor. The following individual involved in review of your submission has agreed to reveal their identity: Alessandro Treves (Reviewer #2).

Essential revisions:

1) Address the reviewer's questions about the causal role of resonance in stabilising grid patterns in this specific continuous attractor model. Provide a clearer and more complete description of the single-neuron dynamics.

2) Substantiate the results with more systematic modelling or mathematical analysis, possibly in a simplified model, to provide intuition or demonstrate the mechanism underpinning the observed stabilisation of grid fields. How specific are these effects to the CAN model architecture and/or grid fields?

*Reviewer #1 (Recommendations for the authors):*

The authors succeed in conveying a clear and concise description of how intrinsic heterogeneity affects continuous attractor models. The main claim, namely that resonant neurons could stabilize grid-cell patterns in medial entorhinal cortex, is striking.

I am intrigued by the use of a nonlinear filter composed of the product of s with its temporal derivative raised to an exponent. Why this particular choice? Or, to be more specific, would a linear bandpass filter not have served the same purpose?

The magnitude spectra are subtracted and then normalized by a sum. I have slight misgivings about the normalization, but I am more worried that , as no specific formula is given, some MATLAB function has been used. What bothers me a bit is that, depending on how the spectrogram/periodogram is computed (in particular, averaged over windows), one would naturally expect lower frequency components to be more variable. But this excess variability at low frequencies is a major point in the paper.

Which brings me to the main thesis of the manuscript: given the observation of how heterogeneities increase the variability in the low temporal frequency components, the way resonant neurons stabilize grid patterns is by suppressing these same low frequency components.

I am not entirely convinced that the observed correlation implies causality. The low temporal frequency spectra are an indirect reflection of the regularity or irregularity of the pattern formation on the network, induced by the fact that there is velocity coupling to the input and hence dynamics on the network. Heterogeneities will distort the pattern on the network, that is true, but it isn't clear how introducing a bandpass property in temporal frequency space affects spatial stability causally.

Put it this way: imagine all neurons were true oscillators, only capable of oscillating at 8 Hz. If they were to synchronize within a bump, one will have the field blinking on and off. Nothing wrong with that, and it might be that such oscillatory pattern formation on the network might be more stable than non-oscillatory pattern formation (perhaps one could even demonstrate this mathematically, for equivalent parameter settings), but this kind of causality is not what is shown in the manuscript.

*Reviewer #2 (Recommendations for the authors):*

I believe in self-organization and NOT in normative recommendations by reviewers: do this, don't do that. Everybody should be able to publish, in some form, what they feel is important for others to know; so I applaud the new open reviews in *eLife*. Besides, this manuscript is written very well, clearly, I would say equanimously, and I do not have other points to raise beyond what I observed in the open review. The figures are attractive, maybe a bit too many and too rich, but clear and engaging. My only suggestion would be, take your band-pass units, and show that they produce grids without any recurrent network. It will be fun.

---

## [Author Response]

Essential revisions:1) Address the reviewer's questions about the causal role of resonance in stabilising grid patterns in this specific continuous attractor model. Provide a clearer and more complete description of the single-neuron dynamics.

To address this and associated concerns about the “brute force amputation of the low

frequencies” that we had employed earlier to construct resonators, we constructed a new mechanistic model for single-neuron resonance that matches the dynamical behavior of physiological resonators. Specifically, we noted that physiological resonance is elicited by a slow activity-dependent negative feedback (Hutcheon and Yarom, 2000). To incorporate resonance into our rate-based model neurons, we mimicked this by introducing a slow negative feedback loop into our single-neuron dynamics (the motivations are elaborated in the new results subsection “Mechanistic model of neuronal intrinsic resonance: Incorporating a slow activity-dependent negative feedback loop”). The single-neuron dynamics of mechanistic resonators were defined as follows:τdsdt=−S−gm(S)+ledmdt=m∞−mτmm∞=(1+exp(S1/2−Sk))−1

Here, 𝑆 governed neuronal activity, 𝑚 defined the feedback state variable, 𝜏 represented the integration time constant, 𝐼_𝑒_ was the external current, and 𝑔 represented feedback strength. The slow kinetics of the negative feedback was controlled by the feedback time constant (𝜏_𝑚_). In order to manifest resonance, 𝜏_𝑚_ > 𝜏 (Hutcheon and Yarom, 2000). The steady-state feedback kernel (𝑚_∞_) of the negative feedback is sigmoidally dependent on the output of the neuron (𝑆), defined by two parameters: half-maximal activity (𝑆_1/2_) and slope (𝑘). The single-neuron dynamics are elaborated in detail in the methods section (new subsection: Mechanistic model for introducing intrinsic resonance in rate-based neurons).

We first demonstrate that the introduction of a slow-negative feedback loop introduce

resonance into single-neuron dynamics (new Figure 9D–E). We performed systematic

sensitivity analyses associated with the parameters of the feedback loop and characterized the dependencies of intrinsic neuronal resonance on model parameters (new Figure 9F–I). We demonstrate that the incorporation of resonance through a negative feedback loop was able to generate grid-patterned activity in the 2D CAN model employed here, with clear dependencies on model parameters (new Figure 10; new Figure 10-Supplements1–2). Next, we incorporated heterogeneities into the network and demonstrated that the introduction of resonance through a negative feedback loop stabilized grid-patterned generation in the heterogeneous 2D CAN model (new Figure 11).

The mechanistic route to introducing resonance allowed us to probe the basis for the

stabilization of grid-patterned activity more thoroughly. Specifically, with physiological

resonators, resonance manifests only when the feedback loop is slow (new Figure 9I;

Hutcheon and Yarom, 2000). This allowed us an additional mechanistic handle to directly probe the role of resonance in stabilizing the grid patterned activity. We assessed the emergence of grid-patterned activity in heterogeneous CAN models constructed with networks constructors with neurons with different 𝜏_𝑚_ values (new Figure 12). Strikingly, we found that when 𝜏_𝑚_ value was small (resulting in fast feedback loops), there was no stabilization of grid-patterned activity in the CAN model, especially with the highest degree of heterogeneities (new Figure 12). With progressive increase in 𝜏#, the patterns stabilized with grid score increasing with 𝜏_𝑚_ =25 ms (new Figure 12) and beyond (new Figure 11B; 𝜏_𝑚_ =75 ms). Finally, our spectral analyses comparing frequency components of homogeneous vs. heterogeneous resonator networks (new Figure panels 13D–F) showed the suppression of low-frequency perturbations in heterogeneous CAN networks.

Together, the central hypothesis in our study was that intrinsic neuronal resonance could stabilize heterogeneous grid-cell networks through targeted suppression of low-frequency perturbations. In the revised manuscript, we present the following lines of evidence in support of this hypothesis (mentioned now in the first paragraph of the Discussion section of the revised manuscript):

1. Neural-circuit heterogeneities destabilized grid-patterned activity generation in a 2D CAN model (Figures 2–3).

2. Neural-circuit heterogeneities predominantly introduced perturbations in the low-frequency components of neural activity (Figure 4).

3. Targeted suppression of low-frequency components through phenomenological (Figure 5C) or through mechanistic (new Figure 9D) resonators resulted in stabilization of the heterogeneous CAN models (Figure 8 and new Figure 11). We note that the stabilization was achieved irrespective of the means employed to suppress low-frequency components: an activity-independent suppression of low-frequencies (Figure 5) or an activity-dependent slow negative feedback loop (new Figure 9).

4. Changing the feedback time constant 𝜏# in mechanistic resonators, without changes to neural gain or the feedback strength allowed us to control the specific range of frequencies that would be suppressed. Our analyses showed that a slow negative feedback loop, which results in targeted suppression of low-frequency components, was essential in stabilizing grid-patterned activity (new Figure 12). As the slow negative feedback loop and the resultant suppression of low frequencies mediates intrinsic resonance, these analyses provide important lines of evidence for the role of targeted suppression of low frequencies in stabilizing grid patterned activity.

5. We demonstrate that the incorporation of phenomenological (Figure 13A–C) or

mechanistic (new Figure panels 13D–F) resonators specifically suppressed lower

frequencies of activity in the 2D CAN model.

6. Finally, the incorporation of resonance through a negative feedback loop allowed us to link our analyses to the well-established role of network motifs involving negative feedback loops in inducing stability and suppressing external/internal noise in engineering and biological systems. We envisage intrinsic neuronal resonance as a cellular-scale activity dependent negative feedback mechanism, a specific instance of a well-established network motif that effectuates stability and suppresses perturbations across different networks (Savageau, 1974; Becskei and Serrano, 2000; Thattai and van Oudenaarden, 2001; Austin et al., 2006; Dublanche et al., 2006; Raj and van Oudenaarden, 2008; Lestas et al., 2010; Cheong et al., 2011; Voliotis et al., 2014). A detailed discussion on this important link to the stabilizing role of this network motif, with appropriate references to the literature is included in the new discussion subsection “Slow negative feedback: Stability, noise suppression, and robustness”.

We thank the reviewers and the editors for their comments, as it allowed us to introduce a physiologically-rooted model for resonance and provide more lines of evidence in support of our hypothesis. We have provided complete descriptions of single neuron dynamics associated with the three kinds of single-neuron models (integrators, phenomenological resonators and mechanistic resonators) in the methods section (equations 6–13), also providing detailed sensitivity analyses with reference to their frequency-dependent response properties (Figure 8 and new Figure 11).

2) Substantiate the results with more systematic modelling or mathematical analysis, possibly in a simplified model, to provide intuition or demonstrate the mechanism underpinning the observed stabilisation of grid fields. How specific are these effects to the CAN model architecture and/or grid fields?

As mentioned above (Q1), in the revised manuscript, we have provided additional systematic modelling and the additional lines of analyses with a physiologically-rooted mechanistic model for resonance (new Figures 9–12, 13D–F; four new Figure supplements). In probing the mechanism behind stabilization, we perturbed the time constant associated with the feedback loop, directly demonstrating the critical role of targeted suppression of neural activity in stabilizing grid-patterned activity in heterogeneous CAN models (new Figure 12). The incorporation of resonance through a negative feedback loop provides further intuition through the well-established role of negative feedback loops in stabilizing systems and reducing the impact of perturbations. This intuition, with appropriate references to engineering and biology literature, is now elaborated in the new discussion subsection “Slow negative feedback: Stability, noise suppression, and robustness”.

With reference to the question on the specificity of these effects to the CAN model architecture and/or grid fields, we submit that our conclusions are limited to the 2D rate-based CAN model presented here (Discussion subsection: “Future directions and considerations in model interpretation”). We postulate that mechanisms that suppress low-frequency components could be a generalized route to stabilize heterogeneous biological networks, based on our analyses and based on the well-established stabilizing role of negative feedback loops. However, we do not claim generalization to other networks or to other models for generating grid-patterned neural activity. As several theoretical frameworks and computational models do not explicitly incorporate the several heterogeneities in afferent, intrinsic and synaptic properties of biological networks, it is essential that the stability of these networks are first assessed in the presence of heterogeneities. The next step would be to assess if there are perturbations in low-frequency components as a consequence of introducing heterogeneities. Finally, if the introduction of heterogeneities destabilized network function and resulted in low-frequency perturbations, then the role of intrinsic resonators in stabilizing these heterogeneities could be probed. As heterogeneities could manifest in different ways, and as their impact on networks could be very different in other networks compared to the network assessed here, we believe that such detailed network- and heterogeneity-specific analyses would be essential before any generalization.

Finally, as our analyses are limited to the stabilizing role of resonating neurons in the

heterogeneous CAN network analyzed here, we have not explored the role of adaptation or resonance in the generation of grid-patterned neural activity. We have however noted with appropriate citations (Discussion subsection: “Future directions and considerations in model interpretation”, first paragraph) that there are other models for the generation of grid-patterned neural activity where resonance has distinct roles compared the stabilizing role proposed here. As the impact of neural heterogeneities have not been assessed in these other models, future studies could assess if heterogeneities could indeed destabilize these models and if resonance could act as a stabilizing mechanism there as well.

Reviewer #1 (Recommendations for the authors):The authors succeed in conveying a clear and concise description of how intrinsic heterogeneity affects continuous attractor models. The main claim, namely that resonant neurons could stabilize grid-cell patterns in medial entorhinal cortex, is striking.

We thank the reviewer for their time and effort in evaluating our manuscript, and for their rigorous evaluation and positive comments on our study.

I am intrigued by the use of a nonlinear filter composed of the product of s with its temporal derivative raised to an exponent. Why this particular choice? Or, to be more specific, would a linear bandpass filter not have served the same purpose?

Please note that the exponent was merely a mechanism to effectively tune the resonance frequency of the resonating neuron. In the revised manuscript, we have introduced a new physiologically rooted means to introduce intrinsic neuronal resonance, thereby confirming that network stabilization achieved was independent of the formulation employed to achieve resonance.

The magnitude spectra are subtracted and then normalized by a sum. I have slight misgivings about the normalization, but I am more worried that , as no specific formula is given, some MATLAB function has been used. What bothers me a bit is that, depending on how the spectrogram/periodogram is computed (in particular, averaged over windows), one would naturally expect lower frequency components to be more variable. But this excess variability at low frequencies is a major point in the paper.

We have now provided the specific formula employed for normalization as equation (16) of the revised manuscript. We have also noted that this was performed to account for potential differences in the maximum value of the homogeneous *vs*. heterogeneous spectra. The details are provided in the Methods subsection “Quantitative analysis of grid cell temporal activity in the spectral domain” of the revised manuscript. Please note that what is computed is the spectra of the entire activity pattern, and not a periodogram or a scalogram. There was no tiling of the time-frequency plane involved, thus eliminating potential roles of variables there on the computation here.

In addition to using variances of normalized differences to quantify spectral distributions, we have also independently employed octave-based analyses (which doesn’t involve normalized differences) to strengthen our claims about the impact of heterogeneities and resonance on different bands of frequency. These octave-based analyses also confirm our conclusions on the impact of heterogeneities and neuronal resonance on low-frequency components.

Finally, we would like to emphasize that spectral computations are the same for different networks, with networks designed in such a way that there was only one component that was different. For instance, in introducing heterogeneities, all other parameters of the network (the specific trajectory, the seed values, the neural and network parameters, the connectivity, etc.) remained *exactly* the same with the only difference introduced being confined to the heterogeneities. Computation of the spectral properties followed *identical* procedures with activity from individual neurons in the two networks, and comparison was with reference to identically placed neurons in the two networks. Together, based on the several routes to quantifying spectral signatures, based on the experimental design involved, and based on the absence of any signal-specific tiling of the time-frequency plane, we argue that the impact of heterogeneities or the resonators on low-frequency components is not an artifact of the analysis procedures.

We thank the reviewer for raising this issue, as it helped us to elaborate on the analysis procedures employed in our study.

Which brings me to the main thesis of the manuscript: given the observation of how heterogeneities increase the variability in the low temporal frequency components, the way resonant neurons stabilize grid patterns is by suppressing these same low frequency components.I am not entirely convinced that the observed correlation implies causality. The low temporal frequeny spectra are an indirect reflection of the regularity or irregularity of the pattern formation on the network, induced by the fact that there is velocity coupling to the input and hence dynamics on the network. Heterogeneities will distort the pattern on the network, that is true, but it isn't clear how introducing a bandpass property in temporal frequency space affects spatial stability causally.Put it this way: imagine all neurons were true oscillators, only capable of oscillating at 8 Hz. If they were to synchronize within a bump, one will have the field blinking on and off. Nothing wrong with that, and it might be that such oscillatory pattern formation on the network might be more stable than non-oscillatory pattern formation (perhaps one could even demonstrate this mathematically, for equivalent parameter settings), but this kind of causality is not what is shown in the manuscript.

The central hypothesis of our study was that intrinsic neuronal resonance could stabilize heterogeneous grid-cell networks through targeted suppression of low-frequency perturbations. In the revised manuscript, we present the following lines of evidence in support of this hypothesis (mentioned now in the first paragraph of the Discussion section of the revised manuscript):

1. Neural-circuit heterogeneities destabilized grid-patterned activity generation in a 2D CAN model (Figures 2–3).

2. Neural-circuit heterogeneities predominantly introduced perturbations in the low frequency components of neural activity (Figure 4).

3. Targeted suppression of low-frequency components through phenomenological (Figure 5C) or through mechanistic (new Figure 9D) resonators resulted in stabilization of the heterogeneous CAN models (Figure 8 and new Figure 11). We note that the stabilization was achieved irrespective of the means employed to suppress low-frequency components: an activity-independent suppression of low-frequencies (Figure 5) or an activity-dependent slow negative feedback loop (new Figure 9).

4. Changing the feedback time constant 𝜏_#_ in mechanistic resonators, *without* changes to neural gain or the feedback strength allowed us to control the specific range of frequencies that would be suppressed. Our analyses showed that a slow negative feedback loop, which results in targeted suppression of low-frequency components, was essential in stabilizing grid-patterned activity (new Figure 12). As the slow negative feedback loop and the resultant suppression of low frequencies *mediates* intrinsic resonance, these analyses provide important lines of evidence for the role of targeted suppression of low frequencies in stabilizing grid patterned activity.

5. We demonstrate that the incorporation of phenomenological (Figure 13A–C) or mechanistic (new Figure panels 13D–F) resonators specifically suppressed lower frequencies of activity in the 2D CAN model.

6. Finally, the incorporation of resonance through a negative feedback loop allowed us to link our analyses to the well-established role of network motifs involving negative feedback loops in inducing stability and suppressing external/internal noise in engineering and biological systems. We envisage intrinsic neuronal resonance as a cellular-scale activity-dependent negative feedback mechanism, a specific instance of a well-established network motif that effectuates stability and suppresses perturbations across different networks (Savageau, 1974; Becskei and Serrano, 2000; Thattai and van Oudenaarden, 2001; Austin et al., 2006; Dublanche et al., 2006; Raj and van Oudenaarden, 2008; Lestas et al., 2010; Cheong et al., 2011; Voliotis et al., 2014). A detailed discussion on this important link to the stabilizing role of this network motif, with appropriate references to the literature is included in the new discussion subsection “Slow negative feedback: Stability, noise suppression, and robustness”.

We thank the reviewer for their detailed comments. These comments helped us to introducing a more physiologically rooted mechanistic form of resonance, where we were able to assess the impact of slow kinetics of negative feedback on network stability, thereby providing more direct lines of evidence for our hypothesis. This also allowed us to link resonance to the well-established stability motif: the negative feedback loop. We also note that our analyses don’t employ resonance as a route to introducing oscillations in the network, but as a means for targeted suppression of low-frequency perturbations through a negative feedback loop. Given the strong quantitative links of negative feedback loops to introducing stability and suppressing the impact of perturbations in engineering applications and biological networks, we envisage intrinsic neuronal resonance as a stability-inducing cellular-scale activity-dependent negative feedback mechanism.

Reviewer #2 (Recommendations for the authors):I believe in self-organization and NOT in normative recommendations by reviewers: do this, don't do that. Everybody should be able to publish, in some form, what they feel is important for others to know; so I applaud the new open reviews in eLife. Besides, this manuscript is written very well, clearly, I would say equanimously, and I do not have other points to raise beyond what I observed in the open review. The figures are attractive, maybe a bit too many and too rich, but clear and engaging. My only suggestion would be, take your band-pass units, and show that they produce grids without any recurrent network. It will be fun.

We thank the reviewer for their detailed and rigorous review, which helped us very much in strengthening the lines of evidence in support of our central conclusions in the manuscript. We thank the reviewer for their belief in self-organization in terms of how the review process should evolve.

We believe that the new resonator model, the demonstration of network stability with the mechanistic resonators and the strong theoretical link to the stability-through-negative feedback literature provides stronger support for our original conclusions. With reference to the grid-cell model without the recurrent network, we agree that it is an important direction to pursue. However, we also respectfully submit that the direction would be a digression from the central hypothesis of our analyses about the stabilizing role of resonating neurons in heterogeneous CAN-based grid-cell networks. We note that addressing the question on grid-patterned activity with individual resonator neurons, in conjunction place-cell inputs and intrinsic/synaptic plasticity, would answer the question of generation of grid-like patterns. However, as our central question relates to heterogeneous networks and stability therein, we instead focused on the reviewer’s comments on our resonator model and built a more realistic model for neural resonators and demonstrated stability with these resonator models.

The generalizability of our conclusions to other grid cells models is an important question, but would require exhaustive analyses of the impact of heterogeneities and resonance on the other models in terms of their ability to generating stable grid-patterned activity. We have stated the limitations of our analyses in the Discussion section, also suggesting directions for future research involving other models for grid-patterned activity generation.

We believe that the new lines of evidence presented in the revised manuscript with the new resonator model, along with the theoretical link to the stability literature involving negative feedbacks have considerably strengthened our central conclusions. We gratefully thank the reviewer for their rigorous and step-by-step review identifying the specific problems in our analyses; this enabled us to rectify these issues in the revised manuscript. We sincerely hope that the revised manuscript is stronger in terms of the lines of evidence presented to support the central conclusions of the overall analyses.

References

Austin DW, Allen MS, McCollum JM, Dar RD, Wilgus JR, Sayler GS, Samatova NF, Cox CD,

Simpson ML (2006) Gene network shaping of inherent noise spectra. Nature 439:608-

611.

Becskei A, Serrano L (2000) Engineering stability in gene networks by autoregulation. Nature 405:590-593.

Burak Y, Fiete IR (2009) Accurate path integration in continuous attractor network models of grid cells. PLoS computational biology 5:e1000291.

Burgess N, O'Keefe J (2011) Models of place and grid cell firing and theta rhythmicity. Current opinion in neurobiology 21:734-744.

Cheong R, Rhee A, Wang CJ, Nemenman I, Levchenko A (2011) Information transduction

capacity of noisy biochemical signaling networks. Science 334:354-358.

Couey JJ, Witoelar A, Zhang SJ, Zheng K, Ye J, Dunn B, Czajkowski R, Moser MB, Moser EI,

Roudi Y, Witter MP (2013) Recurrent inhibitory circuitry as a mechanism for grid formation. Nat Neurosci 16:318-324.

Crawford AC, Fettiplace R (1978) Ringing responses in cochlear hair cells of the turtle

[proceedings]. J Physiol 284:120P-122P.

Crawford AC, Fettiplace R (1981) An electrical tuning mechanism in turtle cochlear hair cells. J Physiol 312:377-412.

Domnisoru C, Kinkhabwala AA, Tank DW (2013) Membrane potential dynamics of grid cells. Nature 495:199-204.

Dublanche Y, Michalodimitrakis K, Kummerer N, Foglierini M, Serrano L (2006) Noise in

transcription negative feedback loops: simulation and experimental analysis. Molecular systems biology 2:41.

Fettiplace R, Fuchs PA (1999) Mechanisms of hair cell tuning. Annual review of physiology 61:809-834.

Giocomo LM, Moser MB, Moser EI (2011a) Computational models of grid cells. Neuron

71:589-603.

Giocomo LM, Zilli EA, Fransen E, Hasselmo ME (2007) Temporal frequency of subthreshold oscillations scales with entorhinal grid cell field spacing. Science 315:1719-1722.

Giocomo LM, Hussaini SA, Zheng F, Kandel ER, Moser MB, Moser EI (2011b) Grid cells use HCN1 channels for spatial scaling. Cell 147:1159-1170.

Hutcheon B, Yarom Y (2000) Resonance, oscillation and the intrinsic frequency preferences of neurons. Trends Neurosci 23:216-222.

Kropff E, Treves A (2008) The emergence of grid cells: Intelligent design or just adaptation? Hippocampus 18:1256-1269.

Lestas I, Vinnicombe G, Paulsson J (2010) Fundamental limits on the suppression of molecular fluctuations. Nature 467:174-178.

Navratilova Z, Giocomo LM, Fellous JM, Hasselmo ME, McNaughton BL (2012) Phase

precession and variable spatial scaling in a periodic attractor map model of medial

entorhinal grid cells with realistic after-spike dynamics. Hippocampus 22:772-789.

Pastoll H, Ramsden HL, Nolan MF (2012) Intrinsic electrophysiological properties of entorhinal cortex stellate cells and their contribution to grid cell firing fields. Frontiers in neural circuits 6:17.

Raj A, van Oudenaarden A (2008) Nature, nurture, or chance: stochastic gene expression and its consequences. Cell 135:216-226.

Savageau MA (1974) Comparison of classical and autogenous systems of regulation in

inducible operons. Nature 252:546-549.

Schmidt-Hieber C, Hausser M (2013) Cellular mechanisms of spatial navigation in the medial entorhinal cortex. Nat Neurosci 16:325-331.

Schmidt-Hieber C, Toleikyte G, Aitchison L, Roth A, Clark BA, Branco T, Hausser M (2017) Active dendritic integration as a mechanism for robust and precise grid cell firing. Nat Neurosci 20:1114-1121.

Stella F, Urdapilleta E, Luo Y, Treves A (2020) Partial coherence and frustration in selforganizing spherical grids. Hippocampus 30:302-313.

Thattai M, van Oudenaarden A (2001) Intrinsic noise in gene regulatory networks. Proc Natl Acad Sci U S A 98:8614-8619.

Tukker JJ, Beed P, Brecht M, Kempter R, Moser EI, Schmitz D (2021) Microcircuits for spatial coding in the medial entorhinal cortex. Physiol Rev.

Urdapilleta E, Si B, Treves A (2017) Selforganization of modular activity of grid cells.

Hippocampus 27:1204-1213.

Voliotis M, Perrett RM, McWilliams C, McArdle CA, Bowsher CG (2014) Information transfer by leaky, heterogeneous, protein kinase signaling systems. Proc Natl Acad Sci U S A 111:E326-333.

Yoon K, Buice MA, Barry C, Hayman R, Burgess N, Fiete IR (2013) Specific evidence of lowdimensional continuous attractor dynamics in grid cells. Nat Neurosci 16:1077-1084.